# Changes in statistical distributions of sub-daily surface temperatures and wind speed

Robert J. H. Dunn[1], Kate M. Willett[1], and David E. Parker[1]

[1]Met Office Hadley Centre, FitzRoy Road, Exeter, EX1 3PB, UK

**Correspondence:** robert.dunn@metoffice.gov.uk

**Abstract.** With the ongoing warming of the globe, it is important to quantify changes in the recent behaviour of extreme events given their impacts on human health, infrastructure and the natural environment. We use the sub-daily, multi-variate, station-based HadISD dataset to study the changes in the statistical distributions of temperature, dewpoint temperature and wind speeds. Firstly, we use zonally averaged quantities to show that the lowest temperatures during both day and night are changing more rapidly than the highest, with the effect more pronounced in the northern high latitudes. Along with increases in the zonally-averaged mean temperature, the standard deviation has decreased and the skew increased (increasing positive tail/decreasing negative tail) over the last 45 years, again with a stronger, more robust signal at higher latitudes. Changes in the distribution of dewpoint temperature are similar to those of temperature. However, changes in the distribution of wind speeds indicate a more rapid change at higher speeds than at lower.

Secondly, to assess in more detail the spatial distribution of changes as well as across seasons and hours of the day we study each station individually. For stations which show clear indications of change in the statistical moments, the higher the statistical moment, generally the more spatially heterogenous the patterns of change. The standard deviations of temperatures are increasing in a band stretching from Europe through to China, but are decreasing across North America and in the high northern latitudes, indicating broadening and narrowing of the distributions respectively. Large seasonal differences are found in the change of standard deviations of temperatures over North America and eastern China. Temperatures in Eastern Asia also have increasing skew in the winter in contrast to the remainder of the year. The dewpoint temperatures show smaller variation in all of the moments, but similar patterns to the temperatures. For wind speeds, apart from the USA, standard deviations are decreasing across the world, indicating a decrease in variability.

Finally, we use quantile regression to show changes in the percentiles of distributions over time. These show an increase of high quantiles of temperature in eastern Europe during the summer, and also in northern Europe for low quantiles in the winter, also indicating broadening and narrowing of the distributions respectively. In North America, the largest changes are at the lower quantiles in northern latitudes for autumn and winter. Quantiles of dewpoint temperature are changing most in the autumn and winter, especially in the northern parts of Europe.

# 1   Introduction

The study of changes in the extremes of essential climate variables is vital given their impacts on human health, infrastructure, agriculture and the natural environment. The Intergovernmental Panel on Climate Change (IPCC) released a Special Report on Extremes (SREX, Field et al., 2012), and therein outlined three simple classes of the way changes in extremes occur; shifting the mean, increasing the variability, and changing the symmetry. Past studies have indicated that there is uncertainty around how changes in the occurrence or intensity of extreme values arise. Are extremes changing due to changes in the location of

the distribution mean with no change in the distribution shape (Griffiths et al., 2005; Simolo et al., 2011), or are changes in the shape of a distribution the primary driver of changes in extremes? If so, is the change in shape only the consequence of a change in the variance or does it also arise from changes in higher order moments (Della-Marta et al., 2007; Ballester et al., 2010)?

There are a whole host of ways to study climate extremes and determine how these have changed over the recent past.

One common approach is to use climate extremes indices to characterise moderate extremes in timeseries of station data, for example those developed by the World Meteorological Organisation (WMO) Commission for Climatology (CCl), World Climate Research Programme (WCRP), and Joint Technical Commission for Oceanography and Marine Meteorology (JCOMM) Expert Team in Climate Change Detection and Indices (ETCCDI, Alexander et al., 2006). A number of datasets comprising these indices now exist, allowing for detailed investigations of past changes and comparisons to model and reanalysis fields

(e.g. Caesar et al., 2006; Donat et al., 2013a, b). Another route uses extreme value theory to model the tails of the distributions, and from these few points characterise the occurrence and intensity of extreme events, including those not yet observed in the modern data record. The "block maximum" approach models a set of e.g. annual, maxima with a Generalised Extreme Value distribution (e.g. Christidis et al., 2011). Alternatively, the "peaks over threshold" approach models all peaks over a fixed threshold with the Generalised Pareto distribution (e.g. Brown et al., 2008).

A further method characterises the complete distribution using all available data to establish what are the causes of changes in climate extremes. The advantage of this approach is that it uses all of the available information, and can be extended to more or less extremal parts of the distribution. A number of the ETCCDI temperature indices use the $10^{\text{th}}$ and $90^{\text{th}}$ percentile values, which only probe moderate extremes. A recalculation of these indices at $1^{\text{st}}$ and $99^{\text{th}}$ percentile values is possible, but given the diverse sources of data in the HadEX datasets (Donat et al., 2013b), this would be a large undertaking by the international

community. Furthermore, as the climate continues to warm, the relative intensity or duration of extreme events against warmer average conditions may be of interest; for example, how warm is the relatively warmest 10 per cent of days now compared to the same fraction selected between 1961-90?

Hence, a number of recent studies have investigated past changes in the distributions of observed land-surface temperatures. Most have used the Global Historical Climate Network Daily (GHCND, Menne et al., 2012) or its gridded version

(HadGHCND, Caesar et al., 2006, temperature only) as these provide daily maximum and minimum temperature values. Donat and Alexander (2012) compared two periods to show that the means of both temperature measures have shifted to warmer values. Along with larger changes in the minimum than the maximum temperatures, this has resulted in increases in the skew

of the distributions. However changes in standard deviation were less significant (at 10%) and more heterogeneous. Over North America, Shen et al. (2011) show that the standard deviation and skewness were decreasing, but that the kurtoses of the maximum temperatures were increasing, in contrast to the decrease in the kurtoses of minimum temperatures. A subsequent study investigated seasonal changes in the distribution moments over 1950-2010 (Cavanaugh and Shen, 2014). McKinnon et al. (2016) showed that for most stations between 1980 and 2015, over just the summer months, trends can mostly be explained by a shift in the distribution with no change in shape. Reanalyses (e.g. Huntingford et al., 2013; Gross et al., 2018), or coupled models (e.g. Lewis and King, 2017) have also been used in the study of changes in the shapes of current and future distributions of essential climate variables (ECVs).

In this study, we use the sub-daily, station data from a single dataset, as opposed to the maximum and minimum daily temperatures (either station based or gridded). We note that by using a single dataset, we have not investigated the effect of dataset choice on our findings, something which can have a large effect (Gross et al., 2018). In order to compare with a number of previous studies we perform three different assessments to see how the distributions of temperature, dewpoint temperature, and wind speed have changed in recent years. We outline the dataset and station selection criteria in Section 2. The three assessments are outlined in Sections 3 to 5, with a summary in Section 6.

## 2 Input Data

To study the changes in distributions related to climate extremes, we use the sub-daily HadISD dataset (Dunn et al., 2012, 2014, 2016; Dunn, 2019). This is updated annually, and now covers a period 1931-2018 inclusive. Improved station selection and merging processes as well as minor changes in the quality control tests are features of version 3.0.0.2018f over version 1.0.x. This version contains 8139 unique station locations, with some of these being composited from individual records in the ISD (Integrated Surface Dataset, Smith et al., 2011). As a result of the long standing issues surrounding the free sharing of observational climate data (Thorne et al., 2017), the bulk of these stations are in the Northern Hemisphere, concentrated in North America and Europe. In contrast to some of the previous studies outlined in the Introduction (e.g. Donat and Alexander, 2012), we do not create a gridded form of the HadISD for this assessment, but retain the individual stations as input to the analyses, similar to e.g. Cavanaugh and Shen (2014); McKinnon et al. (2016).

HadISD undergoes a homogeneity assessment using the Pairwise Homogeneity Algorithm (PHA, Menne and Williams Jr, 2009) to identify the number and size of inhomogeneities in its four main ECVs, including temperature and wind speed (Dunn et al., 2014). This used monthly averages derived from the sub-daily HadISD observations to compare station-pair differences to identify potential jumps in timeseries, a process which is more effective for stations and variables with low variability. Although no adjustments are made, the outputs of the PHA allow stations that have few or small inhomogeneities to be selected, i.e. those with the most homogeneous records (see Dunn et al., 2014). They found that for temperature the best balance of station network against the exclusion of stations with the worst inhomogeneities occurred between $1°$ and $2°$ C. Therefore in all of the investigations below, we select those stations that have fewer than inhomogeneities, allowing jumps of up to $1°$ C (temperature and dewpoint) or $1 \text{ m s}^{-1}$ (as the distribution of the inhomogeneities in the wind speed data are similar,

see Figure 10 of Dunn et al., 2014). This balances retaining sufficient stations to study the behaviour across the globe with removing those with very large inhomogeneities.

As the selected stations still contain uncorrected inhomogeneities, we note that these will have affected the analysis of the statistical properties of the observations outlined herein. But the distributions of the estimated inhomogeneities as shown in Dunn et al. (2014, 2016) do not have a strongly non-zero mean, and the central portions retained by the approach above have no clear asymmetry. Therefore, as the following analyses use combinations of stations or look for contiguous regions of change that cross national (i.e. observing-practice) boundaries, we do not expect large effects arising from the remaining inhomogeneities themselves.

Although the HadISD v3.0.x contains data from 1931 onwards, there is a large increase in the number of stations from 1973 onwards (the reason why HadISD v1.0.x started then, see Fig. 2 in Dunn et al., 2016). Therefore we restrict our analysis to the time period from 1$^{st}$ January 1973 unless otherwise stated. We also remove data from February 29$^{th}$ for ease of analysis.

## 3   Zonal Distribution Changes

We firstly look at the changes in distributions of anomalies of the temperatures, dewpoint temperatures and wind speeds for all stations in a specified latitude band on an annual basis. By combining stations together, the quantity of observations available to define the distribution is increased at the expense of spatial resolution. Donat and Alexander (2012) used the HadGHCND gridded daily temperature dataset (Caesar et al., 2006) to compare two 30-year periods of daily maximum and minimum temperatures (1951-80 and 1981-2010) globally, and for three smaller zonal regions (Northern Hemisphere, Tropics and Southern Hemisphere). Here we develop this method further using the sub-daily data available in the HadISD as well as narrower 10-degree zonal bands.

We select those stations that fall within each 10 degree band. An approximation for the whole-hour timezone of the station is calculated from the station longitude, enabling the Universal Time (UTC) observation time to be converted to local time. We then extract those portions of data that correspond to daytime and night-time, using the time periods 09:00-20:00 and 21:00-08:00 respectively, to ensure that the minimum temperatures are on the whole captured during the night-time portion, and the maximum temperatures in the daytime. At high latitudes these definitions of day- and night-time will vary in accuracy throughout the year, but should split the 24 hour day into the hours with most and least insolation. There are few stations at the highest latitudes where 24-hours of day or night occur.

Climatologies for stations for each day of the year for the day- and night-time data are calculated separately, requiring the equivalent of 15 years of 3-hourly observations over the 1981-2010 period. These climatologies are used to create anomalies from the sub-daily data. The creation of climate anomalies also restricts this analysis to stations that have consistent records within this recent climatology period.

To investigate the changes in the distributions of these zonally averaged quantities over time, we split the data up into nine intervals each of five years in length (1974-78, 1979-83, .., 2014-2018). By not including 1973, we can assess changes over a 45 year period. Furthermore, we only take those stations that have sufficient observations spread over the entire analysis

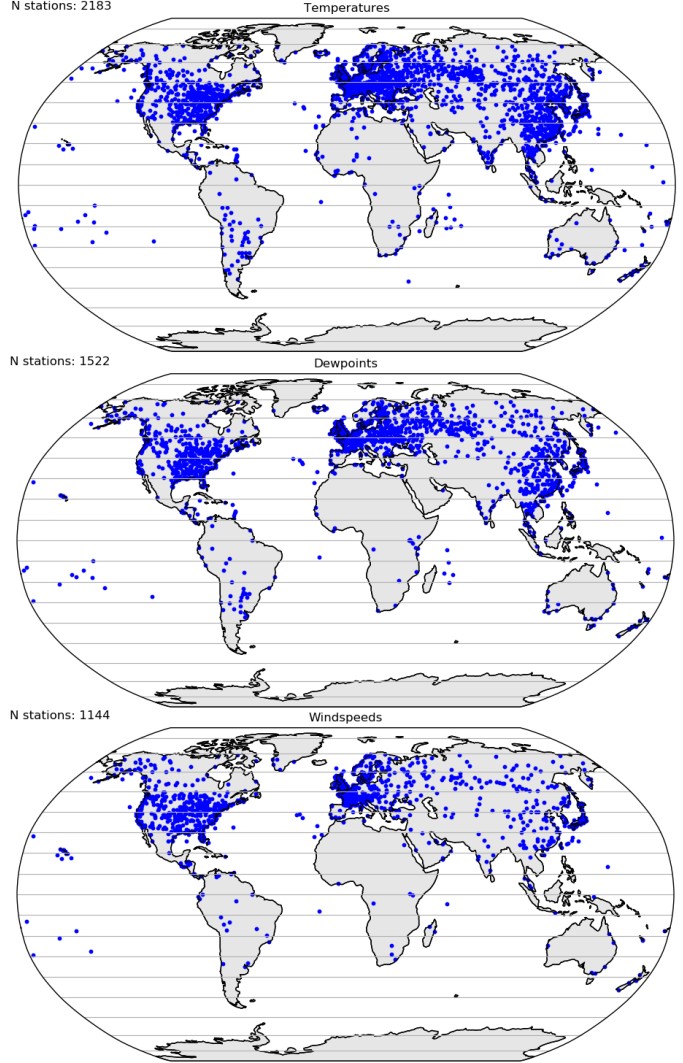

**Figure 1.** Maps showing the locations of HadISD stations contributing to the zonal analysis along with the 10 degree latitude bands for TOP temperature, MIDDLE dewpoints, and BOTTOM windspeeds.

period. In each interval, we require there to be more observations than the number equivalent to hourly observations for 1/4 of the 5-year interval. Also, at least 3/4 of the intervals require sufficient data for the station to be incorporated into the final distribution. This is to ensure that non-uniform distributions of observations within a station record (both within a five-year interval and across the entire record) do not overly influence the final distributions. The stations which pass these selection criteria are shown in Fig 1. A more technical summary of these processing and selection criteria is shown in the Technical Supplement.

We determine the values for the first four statistical moments (mean, standard deviation, skew and [excess] kurtosis) of the resulting distributions. These in turn represent the central tendency, the spread, the asymmetry and the "peaked-ness" of the distribution. Distributions with non-zero skew or kurtosis indicate departures from a pure Gaussian shape. We note that the skew and kurtosis measures are not fully orthogonal (McKinnon et al., 2016), but will use these measures in this analysis because of their well-understood nature. Other statistical moments, e.g. linear (or L-) moments (Hosking, 1990) have also been used for the analysis of changes in the characteristics of distributions (e.g. Fowler et al., 2005; Simolo et al., 2011). However, as noted above, we use the ordinary statistical moments to enable clearer comparison with previous, similar analyses (e.g. Donat and Alexander, 2012).

The linear change in these ordinary moments, along with $1\hat{\sigma}$ range have been derived from the median of pairwise slopes method (e.g. Sen, 1968; Lanzante, 1996), and are shown for each latitude band in Table 1 and Supplementary Information Table S1 & S2 for temperature, dewpoint temperature and wind speed respectively. We do not expect any changes in the parameters of the distributions to be linear, but it is a useful way to summarise their gross changes over time together with the relative certainty of these changes. Latitude bands where the range in trend values does not encompass zero are highlighted in bold, as for these we can be more confident that there is a true change in the value over time.

## 3.1 Temperatures

The results from day- and night-time temperature observations for the Northern Hemisphere are shown in Fig. 2 and Table 1. We can be more confident of changes that are seen in latitude bands containing a large number of stations than in those which only contain a handful. The Southern Hemisphere is shown in Supplementary Information Fig. S1 and Table S1, as it includes considerably fewer stations. It is clearly evident from these figures and tables that there has been a shift in the location of the distributions from the early period to the later period, with the most recent period being on average warmer than the earliest. This behaviour agrees with the results presented in Donat and Alexander (2012) who shows increases in the mean of around 1°C between the early (1951-1980) and late (1981-2010) periods in their study (for all their regions and for both maximum and minimum temperatures). We note that their period of study is longer (60 years) compared to the period presented in this work (45 years). It is also unsurprisingly in agreement with the widespread warming of the Earth's surface (Stocker, 2014).

In both day- and night-time observations, positive increases in the mean that are clearly different from zero are seen in all latitude bands (Table 1). The strongest increases are seen in the highest northern latitudes, with increases on the whole becoming smaller when heading towards the equator for both halves of the day. Changes in the southern hemisphere show less

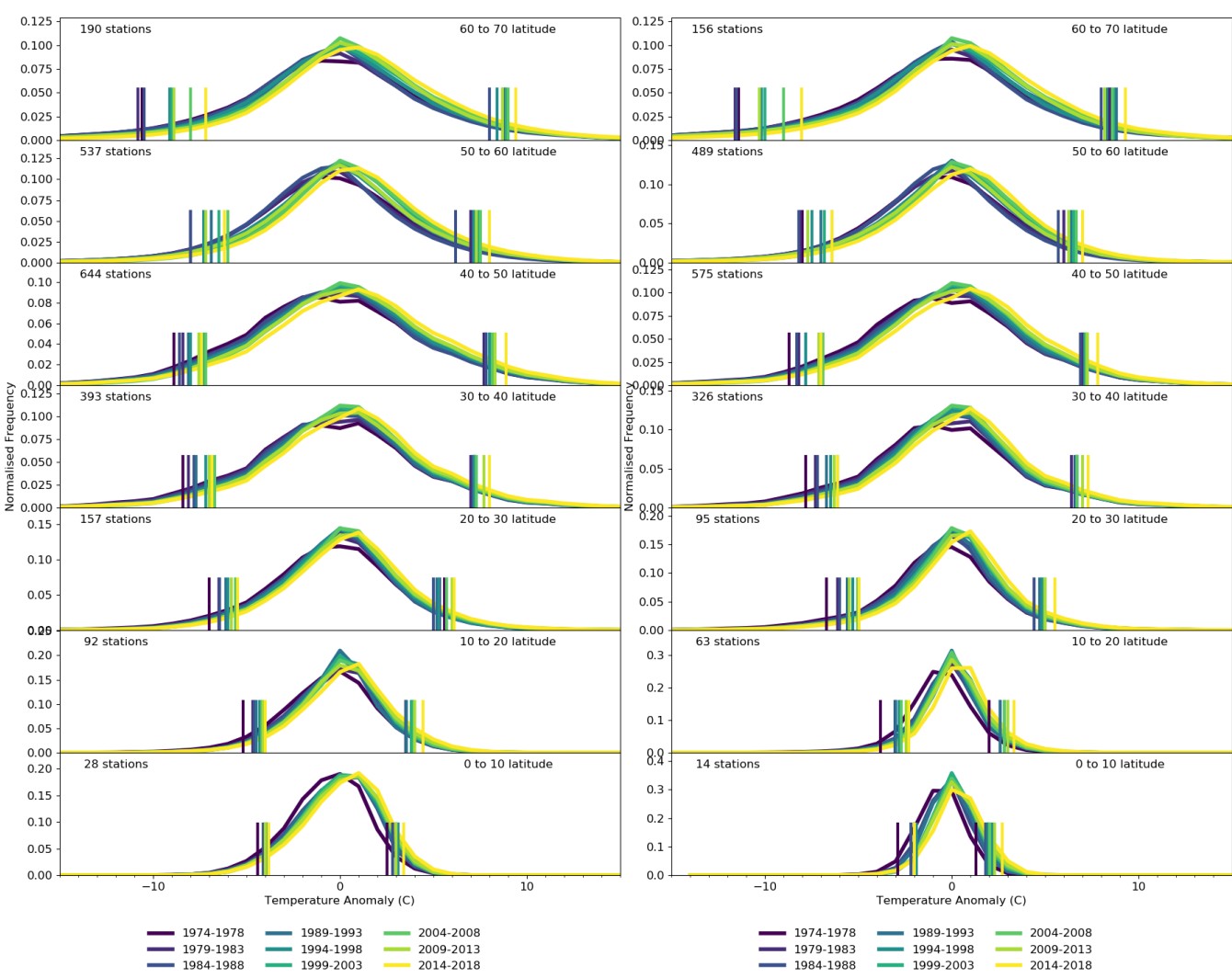

**Figure 2.** Dry bulb temperature distributions in latitudinal bands for LEFT day-, and RIGHT night-time observations for the Northern Hemisphere stations. The $y$-axis has been normalised for ease of comparison. The number of stations contributing to each set of distributions is shown in the top left of each sub-panel, with the latitude range in the top right. The short vertical bars indicate the location of the $5^{th}$ and $95^{th}$ percentile values for each distribution.

variation with latitude, and are on the whole smaller than in the north, though still showing overall increases. There are few differences in the day- and night-time increases for each latitude band, in most cases the ranges overlap.

The northern mid-to-high latitudes also see a consistent decrease in the standard deviations in both day and night time. However, south of 20°N, the signal is less consistent. In fact, during day- and night-time most tropical and southern-latitude bands have ranges that encompass zero, though most bands suggest decreases. Although Donat and Alexander (2012) also found decreases in standard deviation in the extra-tropics, between 30°S-30°N they found an increase when using zonal averages. This decrease suggests that the maximum temperatures are warming less quickly than the minimum temperatures, reducing the spread of the distribution in the northern hemisphere.

Consistent changes in the skew and kurtosis for both day- and night-time or across latitude bands are not as apparent in Table 1. Slight increases in skew (growth in the positive tail and shrinking of the negative tail) are seen for the daytime high latitudes (north and south). In contrast, during night-time, some tropical regions show decreases in skew. The kurtosis decreases in the tropics at both times of day, with a more mixed signal at the high latitudes. Donat and Alexander (2012) also found an increase in the skew in all regions except for $T_{max}$ in the tropics, which is broadly consistent with the changes shown in Table 1.

### 3.2 Dewpoint Temperatures

The changes in the dewpoint temperatures (Supplementary Information Table S1 and Figs. S2-S3) show a slightly more complex picture than for the dry-bulb temperatures. In general, the mean again shows an increase in both day- and night-time values, with larger increases over time in the northern hemisphere high-latitudes, similar to the dry-bulb temperatures. In the southern hemisphere however, and especially at high latitudes, this increasing signal is less strong with few cases where the confidence intervals do not span zero. One southern latitude band even shows a clear decrease in mean dewpoint temperatures over the last 45 years. Also, trends close to the equator are greater than at latitudes immediately to the north and south of this equatorial belt.

The tendency has been for standard deviations in many northern latitudes to decrease, but there are also a number of increases, especially in the southern hemisphere, though determined from few stations. For the skew, in the daytime, these increase in the north and decrease in the south, with a similar, but not as consistent pattern in the night-time. The changes in kurtosis do not show any clear pattern in comparison to the other moments.

### 3.3 Wind speeds

We also assess changes in the sub-daily wind speeds (for example, 1 or 2 minute average values over the USA DeGaetano, 1997, or 10 minute average values over the UK), but there are fewer stations where this variable has been sufficiently observed (see Supplementary Information Table S2 and Fig. S4). In fact, all latitude bands in the Southern Hemisphere have fewer than ten stations, and so we do not study this hemisphere. We use the same method as for the temperatures but note that the anomalies will be truncated at the negative end by the zero-bounded nature of this variable.

For the northern hemisphere, most latitude bands show reductions in the mean and also the standard deviation over time for both day- and night-time observations, with an uncertainty range distinct from zero. In the lower mid-latitudes, there are

| Band | N-stations | Mean (°C decade$^{-1}$) | St. Dev. (°C decade$^{-1}$) | Skew (decade$^{-1}$) | Kurtosis (decade$^{-1}$) |
|---|---|---|---|---|---|
| | | | Day | | |
| $70 \rightarrow 60$ | 190 | **0.40 (0.36 → 0.49)** | **−0.19 (−0.22 → −0.15)** | **0.04 (0.03 → 0.06)** | 0.06 (−0.00 → 0.14) |
| $60 \rightarrow 50$ | 537 | **0.33 (0.29 → 0.42)** | **−0.09 (−0.13 → −0.04)** | **0.03 (0.02 → 0.05)** | −0.04 (−0.14 → 0.04) |
| $50 \rightarrow 40$ | 644 | **0.33 (0.32 → 0.39)** | **−0.06 (−0.09 → −0.03)** | −0.00 (−0.01 → 0.03) | −0.02 (−0.09 → 0.00) |
| $40 \rightarrow 30$ | 393 | **0.29 (0.27 → 0.30)** | **−0.10 (−0.12 → −0.04)** | **0.02 (0.01 → 0.04)** | **0.05 (0.03 → 0.09)** |
| $30 \rightarrow 20$ | 157 | **0.24 (0.23 → 0.27)** | **−0.05 (−0.07 → −0.01)** | **0.03 (0.02 → 0.06)** | **0.06 (0.05 → 0.09)** |
| $20 \rightarrow 10$ | 92 | **0.18 (0.16 → 0.22)** | −0.01 (−0.03 → 0.03) | **0.01 (0.00 → 0.04)** | 0.01 (−0.02 → 0.07) |
| $10 \rightarrow 0$ | 28 | **0.16 (0.13 → 0.19)** | 0.00 (−0.00 → 0.01) | −0.01 (−0.03 → 0.01) | **−0.10 (−0.11 → −0.07)** |
| $0 \rightarrow -10$ | 12 | **0.18 (0.17 → 0.24)** | −0.02 (−0.02 → 0.01) | **0.04 (0.03 → 0.06)** | **−0.15 (−0.16 → −0.13)** |
| $-10 \rightarrow -20$ | 32 | **0.15 (0.13 → 0.19)** | **0.04 (0.03 → 0.07)** | 0.01 (−0.00 → 0.04) | **−0.17 (−0.20 → −0.06)** |
| $-20 \rightarrow -30$ | 37 | **0.12 (0.11 → 0.15)** | **−0.08 (−0.09 → −0.07)** | **0.03 (0.01 → 0.05)** | **0.20 (0.17 → 0.22)** |
| $-30 \rightarrow -40$ | 32 | **0.21 (0.18 → 0.26)** | 0.01 (−0.03 → 0.04) | **0.07 (0.06 → 0.09)** | **0.11 (0.09 → 0.16)** |
| $-40 \rightarrow -50$ | 11 | **0.09 (0.07 → 0.19)** | 0.01 (−0.02 → 0.05) | **0.05 (0.04 → 0.08)** | −0.03 (−0.05 → 0.01) |
| | | | Night | | |
| $70 \rightarrow 60$ | 156 | **0.39 (0.33 → 0.48)** | **−0.16 (−0.20 → −0.14)** | 0.00 (−0.01 → 0.03) | 0.04 (−0.03 → 0.12) |
| $60 \rightarrow 50$ | 489 | **0.30 (0.25 → 0.40)** | **−0.07 (−0.10 → −0.03)** | **0.03 (0.01 → 0.06)** | −0.10 (−0.19 → 0.00) |
| $50 \rightarrow 40$ | 575 | **0.32 (0.30 → 0.35)** | **−0.10 (−0.11 → −0.07)** | 0.01 (−0.01 → 0.04) | −0.01 (−0.07 → 0.04) |
| $40 \rightarrow 30$ | 326 | **0.29 (0.27 → 0.30)** | **−0.11 (−0.13 → −0.04)** | **0.05 (0.03 → 0.07)** | 0.04 (−0.01 → 0.12) |
| $30 \rightarrow 20$ | 95 | **0.27 (0.25 → 0.29)** | **−0.06 (−0.08 → −0.00)** | 0.00 (−0.01 → 0.02) | **0.11 (0.09 → 0.14)** |
| $20 \rightarrow 10$ | 63 | **0.19 (0.19 → 0.26)** | **−0.05 (−0.05 → −0.02)** | **0.07 (0.06 → 0.10)** | **−0.29 (−0.40 → −0.20)** |
| $10 \rightarrow 0$ | 14 | **0.22 (0.19 → 0.24)** | 0.00 (−0.00 → 0.03) | **−0.07 (−0.08 → −0.05)** | **−0.42 (−0.49 → −0.38)** |
| $0 \rightarrow -10$ | 6 | **0.21 (0.16 → 0.26)** | **−0.07 (−0.08 → −0.03)** | **0.02 (0.00 → 0.06)** | **−0.55 (−0.71 → −0.32)** |
| $-10 \rightarrow -20$ | 8 | **0.24 (0.20 → 0.28)** | −0.02 (−0.03 → 0.02) | **−0.10 (−0.12 → −0.01)** | **0.21 (0.09 → 0.32)** |
| $-20 \rightarrow -30$ | 21 | **0.17 (0.13 → 0.20)** | **−0.05 (−0.07 → −0.02)** | **0.02 (0.01 → 0.05)** | **0.10 (0.05 → 0.12)** |
| $-30 \rightarrow -40$ | 24 | **0.22 (0.21 → 0.23)** | **−0.07 (−0.08 → −0.01)** | **0.04 (0.04 → 0.06)** | **0.14 (0.11 → 0.18)** |
| $-40 \rightarrow -50$ | 11 | **0.09 (0.06 → 0.16)** | **0.01 (0.00 → 0.05)** | **0.01 (0.00 → 0.03)** | **−0.08 (−0.11 → −0.02)** |

**Table 1.** Linear change per decade in fits to parameters in zonal analysis for TOP day-, and BOTTOM night-time temperature observations over 1974-2018. Values in bold show parameters and bands where the $1\sigma$ range of the fitted trend does not include zero.

indications for a small increase in standard deviations, but at a lower magnitude than other changes. Tropical regions show decreases in the skewness over day- and night-time, but high latitudes an increase during the day.

These decreases in the mean wind speed over land have been noted in past studies (termed "stilling", Roderick et al., 2007; McVicar et al., 2012), and monitoring studies have also shown decreases over time for the high and low wind speed values (e.g. Azorin-Molina et al., 2019). However, these recent regular global assessments also use the HadISD and so differences are not

expected. A number of possible explanations have been proposed to explain the stilling, which may be reversing in recent years (Azorin-Molina et al., 2019), including amongst others changes in the surface roughness because of land use changes (Vautard et al., 2010), variability of the atmospheric circulation (Azorin-Molina et al., 2016), and instrumental issues with wind sensors (Azorin-Molina et al., 2017, 2018).

### 3.4   Zonal Summary

As can be seen in both Fig. 2 and Table 1 (as well as Fig. S1 in the Supplementary Information) there is a clear increase in the mean temperatures with time across all latitude bands and times of day. However, this is also combined with a decrease in the standard deviation over the northern hemisphere. The width of the distribution of temperatures is therefore narrowing. Also, the skew of the distribution is becoming more positive, which indicates that the positive tail is growing and the negative tail is shrinking. Together, these three changes indicate that there has been a greater change in the low tail than in the high tail of

the temperature distribution. The changes in the kurtosis are mainly increases (the tropical decreases rely on small samples), indicating a change to more peaked distributions, and a reduction in both tails. Although Donat and Alexander (2012) found similar increases in the mean and skew of daily maxima and minima, they showed increases in the standard deviation for tropical regions not found here.

Markers indicating the location of the $5^{th}$ and $95^{th}$ percentile values in Fig. 2 track the changing tails of the distributions.

Qualitatively for the northern hemisphere, the low tails have changed more than the high tails, as the vertical lines cover a greater spread, and more so for the higher latitudes than the tropical regions. These highlight changes in the standard deviation and also the skew on top of changes in the mean. However, there is no striking difference between the day- and night-time plots for temperature. The number of stations in the southern hemisphere is much smaller, but there are no systematic differences between the behaviours of the low and high tails (see Supplementary Information). A similar pattern is seen for the dewpoint

temperatures, with a greater shift for the lower tail, and no striking difference between night- and daytime values. The wind speeds show little change in the lower tails (Supplementary Information Fig. S4), but decrease in the value of the $95^{th}$ percentile over time. Further investigations into the changes in the $5^{th}$ and $95^{th}$ percentile values are presented in Section 5.

The intention of investigating the day and night time values separately was to echo analyses conducted in other studies of daily maximum and minimum temperatures. Many of these show that the minimum temperatures are changing faster than the

maxima, and so we expected to find a difference between day and night time, either for the parameters of the distributions or in the tail markers. However, the clearest difference remains between the upper and lower tails. This analysis shows increases in the mean and skew, and decreases in the northern hemisphere standard deviations. These changes in the statistical moments of the temperature distributions indicate that the lower tails are shifting more rapidly than the upper tails. Hence, as well as a

change in the location of the distribution, the shape has also changed, resulting in changes to occurrence of extremes. Although fewer stations contribute to the wind speed observations, these also show changes in location and shape, with larger changes in the upper tails caused by decreases in the mean and standard deviations.

Increases in the mean for the temperature and dewpoint temperatures are expected under a warming world (Stocker, 2014), and so it is the coherent changes in the higher moments which are of greater interest in this analysis. A greater change in the minima than the maxima can drive both the decrease in standard deviation and also the increases in the skew, but interestingly we see no difference between the magnitude of night-time changes compared to daytime ones. Some studies using climate models suggest that changes in extremes in a warming world (from anthropogenic forcings) come from changes in the shape as well as location of the temperature distributions (Clark et al., 2006; Hegerl et al., 2004). More recent work has looked at the effect of moisture on extreme temperatures, in the soil (Whan et al., 2015) or the atmosphere (Sherwood and Huber, 2010; Matthews, 2018), where the separation into latent and sensible heating when moisture is present can reduce the peak temperatures.

We note that observations in HadISD have a finite numerical resolution, arising from the point of recording, through transcription, rounding, conversion and truncation, and in cases combinations and multiples of these. As in this part of our analysis we blend anomalies calculated using daily climatologies from different stations together, and assuming a random distribution of these effects within a latitude band we do not discuss the impact of all these processes at this point, but we do so in the next section (Section 4).

This analysis combines station anomalies together in zonal bands, separately for local day- and night-time. In doing so, it removes any small scale, regional changes in the characteristics of the distributions of the meteorological variables. We have also not split the analysis up into seasons (neither 3-month or wet/dry), and so large changes occurring in only part of the year will have been diluted in this analysis. In the following section we improve the assessment of regional and seasonal changes but in doing so reduce the number of observations available to characterise the distributions.

## 4   Station Distribution Changes

To investigate the geographical patterns of changes in these meteorological parameters in more detail than in Section 3, we study each station individually. This reduces the amount of data available to characterise the distribution at the gain of increased spatial information. A similar approach was used by e.g. Cavanaugh and Shen (2014); McKinnon et al. (2016) in their investigations into the changing shape of temperature distributions on a daily basis from GHCND. However, the sub-daily nature of HadISD means we are able to investigate the distributions at different times throughout the day, rather than the changes of the maximum and minimum values independently. A large proportion of the stations in the HadISD only report every three hours, rather than hourly. Therefore we use the station longitude to adjust from UTC to local time, collating to the nearest 3-hourly time point (00:00, 03:00, ...).

For each station a climatology over 1981-2010 is calculated for each day of the year for each 3-hourly time point ($365 \times (24/3)$), requiring that at least 15 years of data are present. These 3-hourly climatologies are subtracted from the observations to

create 3-hourly anomalies. The analysis is performed using observations over the entire year and also for standard three-month seasons.

The 45 years (1974-2018 inclusive) are split into five 9-year periods, and the anomalies within each period and at each three-hourly time point (0000, 0300, ... etc.), are combined, annually as well as seasonally. Using fewer periods of longer length than in Section 3 allows for better characterisation of the distributions at the individual station level, at the expense of the number of periods. This is in contrast to the previous analysis (Section 3) where combining stations together increases the number of observations contributing to a distribution. For the three-month seasons, using periods nine years in length means there are maximally around 800 observations from which the distribution at a single time point can be characterised, if all observations are present (3 months $\times \sim 30$ days $\times$ 9 years). The first four moments of the distributions are only calculated if there are more than 300 observations present within a 9-year period (roughly the equivalent of 1 months worth of observations at a given time point). A more technical summary of these processing and selection criteria is shown in the Technical Supplement.

We again use the median of pairwise slopes method (Sen, 1968; Lanzante, 1996) to characterise the change in the moments over time, but only if there are sufficient data in at least four of the five periods. Again, we do not expect any changes to be linear, but this enables the convenient visual inspection of changes over time. Even if our data completeness criteria result in the inclusion of stations which have a non-uniform distribution of observations across the years and seasons, coherent, robust changes (if they are present) will still stand out across the stations.

## 4.1 Temperature

In Fig. 3 we show changes in the mean, standard deviation, skew and kurtosis for the temperature data for all seasons from HadISD at 15:00 and at 06:00 local time in Fig. 4, to correspond approximately with the times of maximum/daytime and minimum/nighttime temperature respectively. Other times are available in the Supplementary Material. Despite 8139 stations being available in the HadISD v3.0.0.2018f dataset, after selection for homogeneity and length of record, only around 3200 have quantified changes in distributional parameters, the total number of stations is shown in parentheses at the bottom right-hand corner of each panel. Those stations where the $1\hat{\sigma}$ range of the fitted trend excludes zero are plotted with a larger symbol than those where this range encompasses zero. As is also clear from these figures, there are many fewer stations in the southern hemisphere than in the north. The highest concentration of stations is in Europe and eastern Asia, and this uneven meridional distribution will have affected the zonal average calculations in Section 3.

For the distributions of daytime/maximum temperature (15:00, Fig. 3), it is immediately apparent that there are large and coherent changes in the mean and standard deviation over the period of record of these stations. Almost all changes in the mean are positive, with the strongest signal in Eurasia, and lower values in North America. Approximately 95 per cent (3062 plotted out of a total of 3224) of the stations have trends where the uncertainty range does not encompass zero. The majority of changes in standard deviation at high northern latitudes are negative, along with most of central North America. There are increases over Europe and parts of China and southern Russia. Fewer stations (around 45 per cent, 1542/3224) have trends where the uncertainty range of the trends do not encompass zero.

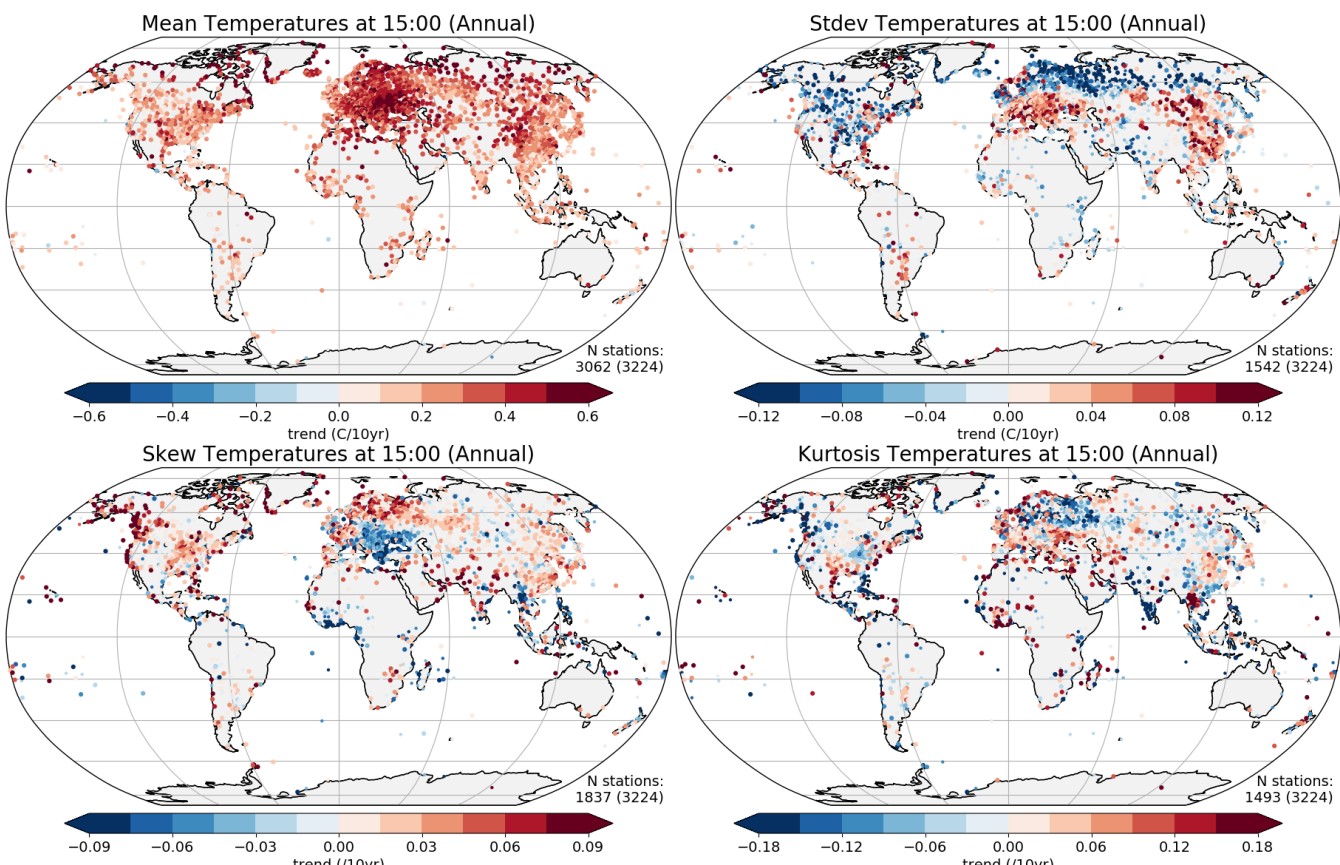

**Figure 3.** Trend over 1974-2018 in distribution parameters for temperature at 15:00 local time, showing the mean, standard-deviation, skew and kurtosis. The stations where the $1\hat{\sigma}$ range of the fitted trend excludes zero are plotted with a larger symbol (the number of these is shown in the bottom right hand corner of each plot). The total number of stations available is shown in parentheses.

Changes in skew and kurtosis have a more heterogenous signal, and again around half the stations exhibit changes where the uncertainty range does not include zero. However, the changes are not completely random, with consistent regional patterns in some areas, but these are smaller than for the standard deviation. The general pattern is for slight increases in the skew, except in a belt around the tropics and in south-east Europe. The kurtosis shows decreases in north-central Europe, and a band extending eastwards from the White Sea across to China and India, with a more mixed signal elsewhere with smaller areas

showing coherent changes.

The night time observations (06:00) show very similar general patterns for changes in the mean and standard deviation, with quasi-global increases in the mean, and decreases in the standard deviation at high latitudes and increases for China, Europe and south-eastern Russia. However, the magnitude of the changes in the mean are lower than for the daytime (15:00) observation, especially over Europe.

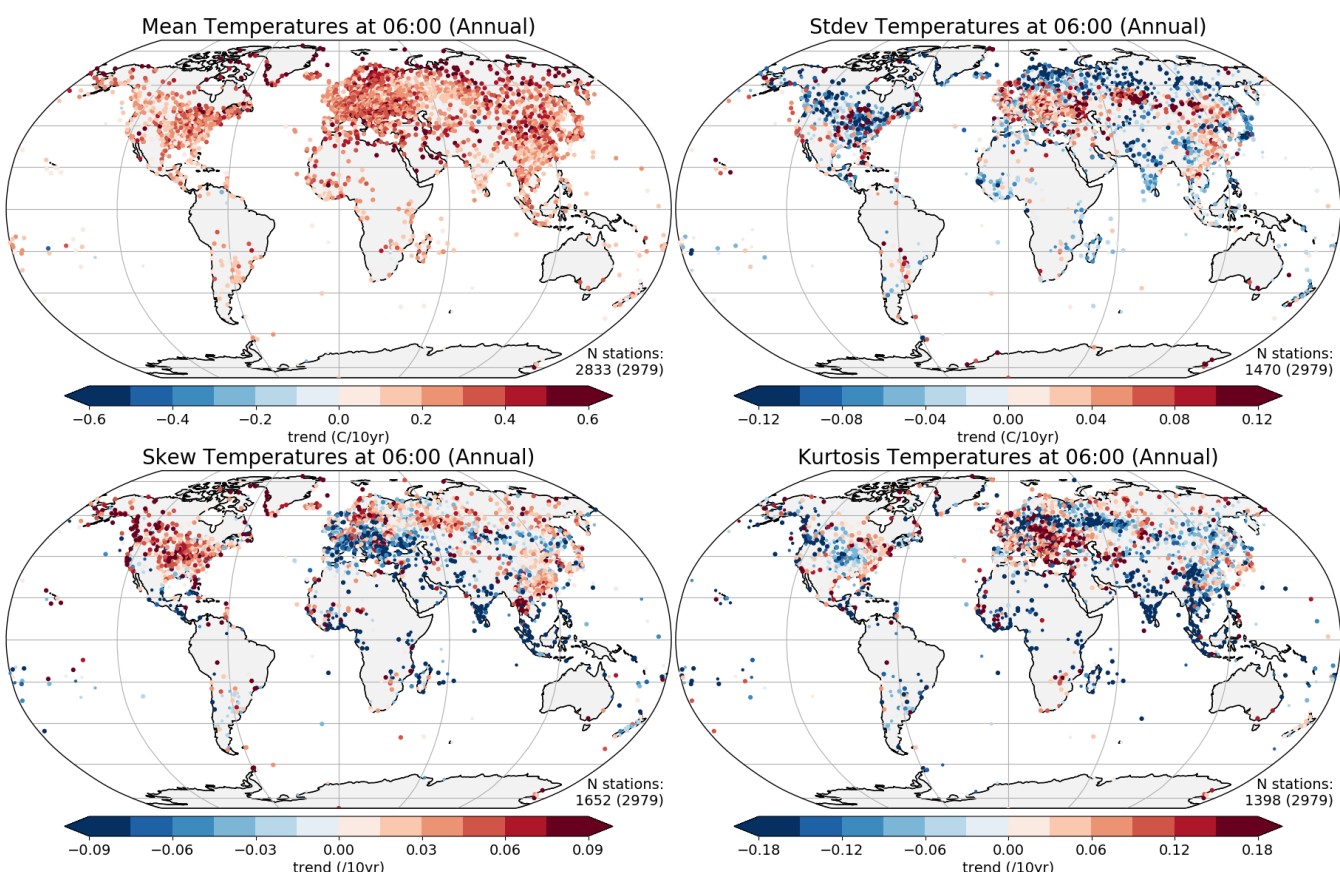

**Figure 4.** Trend over 1974-2018 in distribution parameters for temperature at 06:00 local time, showing the mean (°C), standard-deviation (°C), skew and kurtosis. The stations where the $1\hat{\sigma}$ range of the fitted trend excludes zero are plotted with a larger symbol (the number of these is shown in the bottom right hand corner of each plot). The total number of stations available is shown in parentheses.

There is a stronger change between the skewness for the 06:00 observations and those from 15:00, with larger regions showing a coherent signal. Decreases are visible more widely in Europe, and also in India and parts of south-east Asia; and increases in the high latitudes, as well as North America. Regions with a strong decrease in kurtosis are mid-western North America, India through China and a band in north western Russia, but in contrast to 15:00, large parts of Europe show an increase.

### 4.1.1  Temperature Changes Across the Day

In Fig. 5 we show the maps for changes in the mean for all eight of the 3-hourly time-stamps. All hours of the day show an increase in the mean, with the strongest signals during the daytime (09:00, 12:00, 15:00). The region with the largest trends is Europe. However, it also has the highest station densities, and so the eye is more drawn to this region than areas which also show large trends but from a sparser station network.

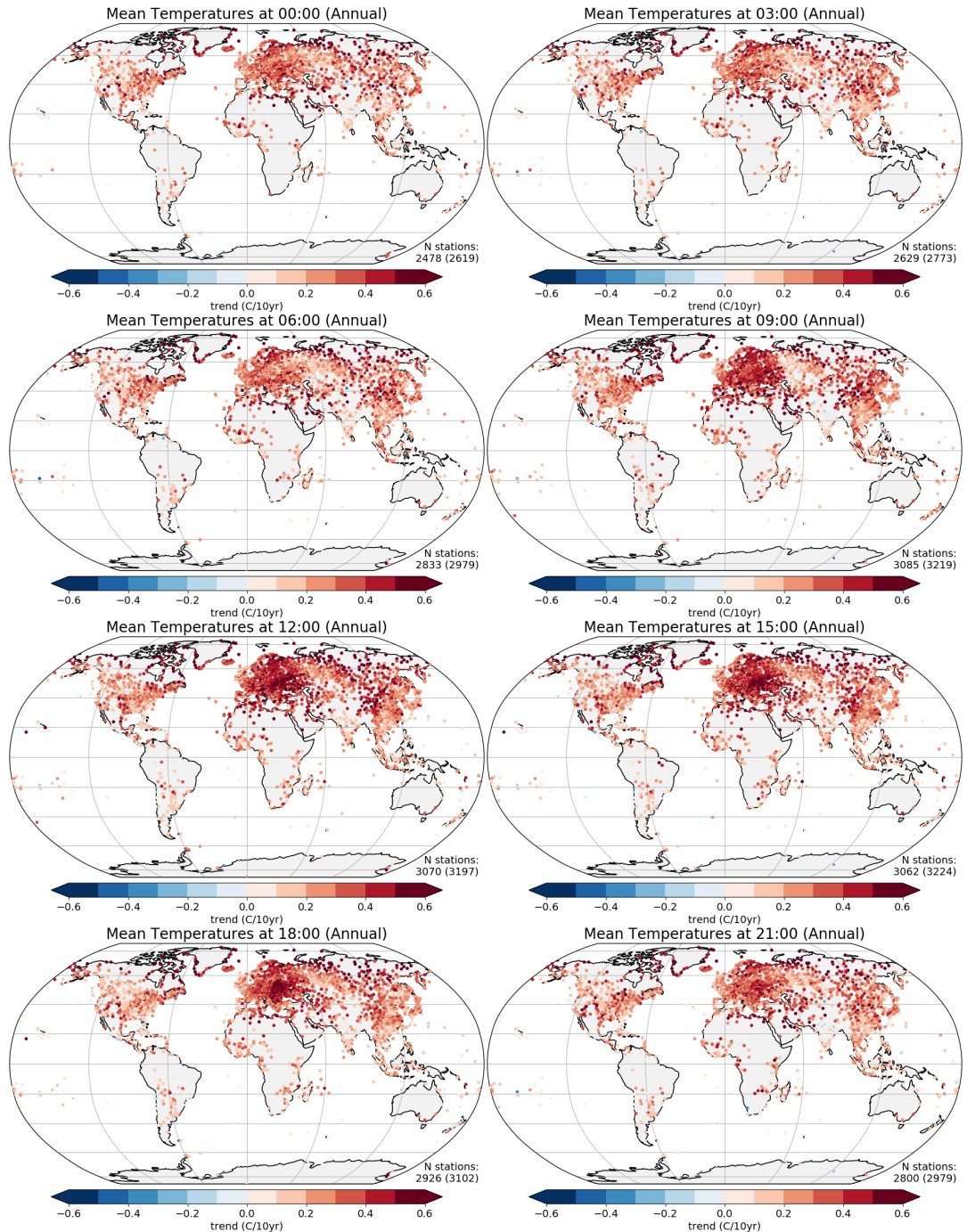

**Figure 5.** Trend over 1974-2018 in distribution mean for temperature at each three hourly interval. The stations where the $1\hat{\sigma}$ range of the fitted trend excludes zero are plotted with a larger symbol (the number of these is shown in the bottom right hand corner of each plot). The total number of stations available is shown in parentheses.

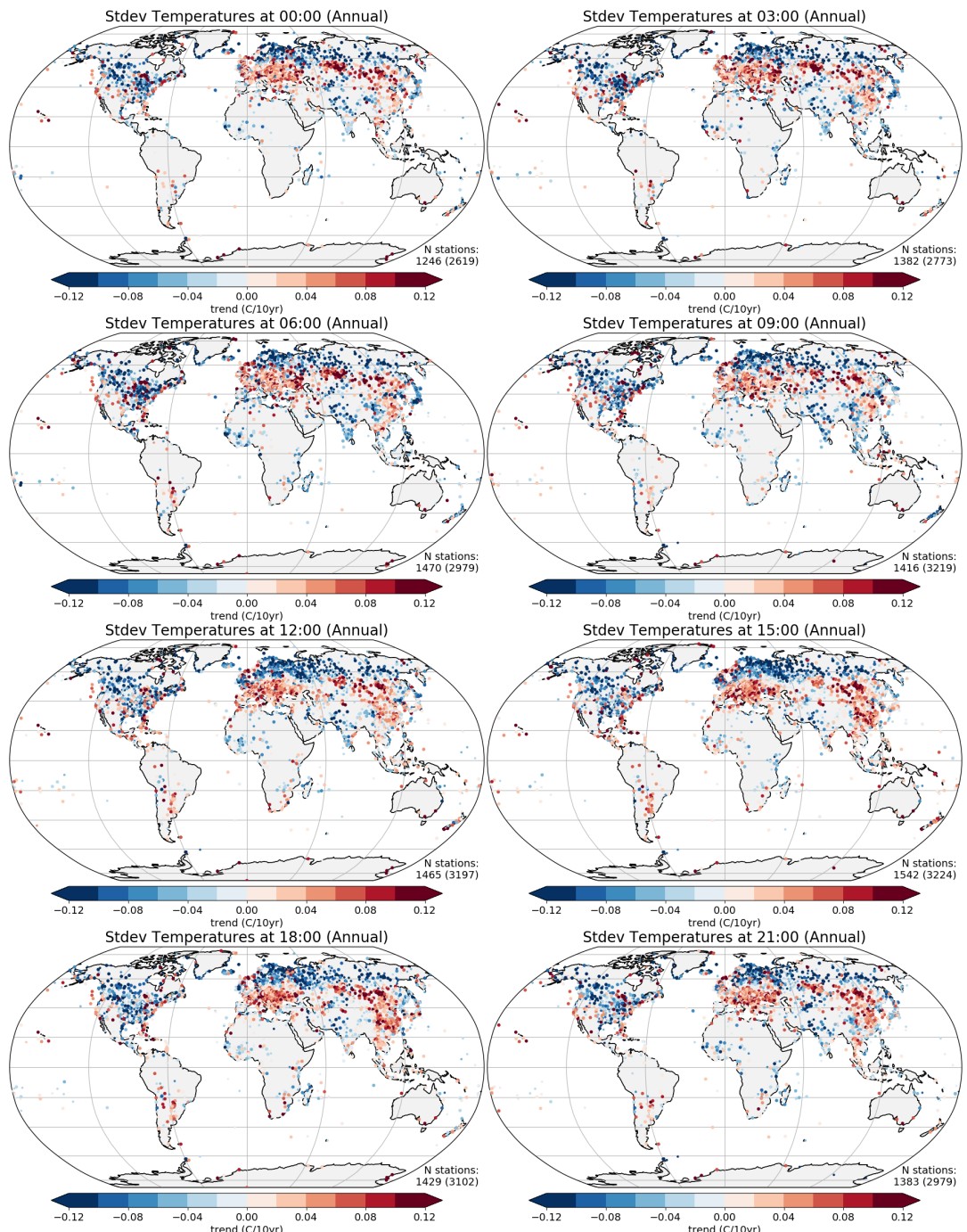

**Figure 6.** Trend over 1974-2018 in distribution standard deviation for temperature at each three hourly interval. The stations where the $1\hat{\sigma}$ range of the fitted trend excludes zero are plotted with a larger symbol (the number of these is shown in the bottom right hand corner of each plot). The total number of stations available is shown in parentheses.

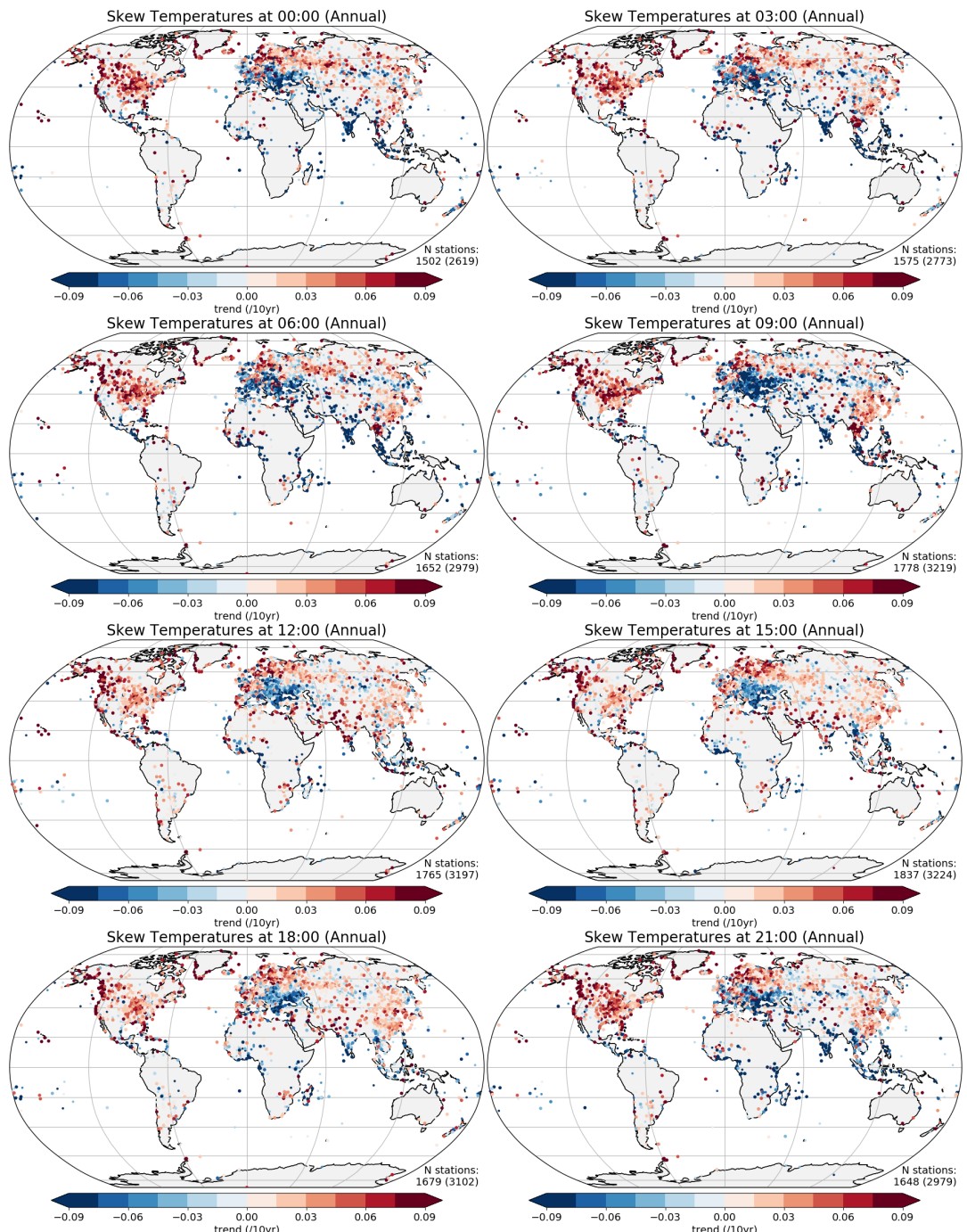

**Figure 7.** Trend over 1974-2018 in distribution skewness for temperature at each three hourly interval. The stations where the $1\hat{\sigma}$ range of the fitted trend excludes zero are plotted with a larger symbol (the number of these is shown in the bottom right hand corner of each plot). The total number of stations available is shown in parentheses.

The trends in standard deviation show decreases at all hours in high northern latitudes, North America, and southern Asia (Indian subcontinent and surroundings), Fig. 6. Increases in the standard deviation are seen in a band, from Europe across to China. The skew from HadISD (Fig. 7) shows increases in most regions except the majority of Europe and parts of Asia. The decreases in skew over Europe are strongest in the mid-morning (09:00) and weakest in the late evening (21:00). Also, changes in standard deviation and skew are the inverse of each other, when comparing Figs. 6 and 7.

### 4.1.2  Seasonal Temperature Changes

In the Supplementary Information Figs. S9-S12 we show the changes for the first four moments from the 15:00 observations across the four seasons of the year (and Figs. S13-S16 for the 06:00 observations), with the standard deviations at 15:00 being reproduced here in Fig 8. Again, the strongest variations are over Eurasia. Changes in the mean temperature at 15:00 are smallest in boreal winter[1] (DJF), and strongest over a wide area in spring (MAM). However, summer (JJA) temperatures 320    increase strongly with time around the Mediterranean and across continental Europe, and winter (DJF) has strong warming in the western part of Russia and Scandinavia. In North America, the largest trends are seen in SON (especially in higher latitudes), with little variation over the other seasons, especially spring and summer. A similar, but less intense pattern is observed at 06:00.

Although increases in the standard deviation are seen across Asia in springtime and summer at 15:00, a more mixed pattern 325    is present in this region in autumn; decreases are seen in the far east in winter (Fig 8). At 0600, the pattern is somewhat similar, though with the Indian sub-continent and surrounding regions showing consistent decreases throughout the year. Over North America, standard deviations increase during spring, but mostly decrease during the other seasons, except in summer at 06:00. Further increases in the standard deviation are also found around the Mediterranean and in south-eastern Europe for most of the year, with decreases in the north of the continent (British Isles and Scandinavia). Increases in the standard deviation are 330    generally weaker and decreases more prevalent across all seasons at 06:00 then at 15:00.

Patterns of changes in the skew have some clear features, with decreases in China for all seasons except winter. Europe also shows decreases in the spring, but a more mixed pattern for the rest of the year. Increases in the skew at 15:00 are found in northern high latitudes during spring, as well as India for most of the year. Over North America, decreases in skew are observed during spring and autumn in the southern half, but in the northern half during summer.

### 4.1.3  Temperature Discussion

The results at 15:00 and 06:00 can be compared to the maximum and minimum temperatures in the study by Donat and Alexander (2012) using HadGHCND between 1951 and 2010. Unsurprisingly, the increase in the mean temperatures is also visible in the HadGHCND study, along with numerous other studies on recent climate change. However, HadGHCND shows stronger increases in the minimum temperatures (analogous to 06:00 here), whereas HadISD shows stronger increases in daytime (15:00)

---

[1]We will use the seasonal descriptions for the northern mid-latitudes, as the station density is highest and we are able to draw the clearest inferences. We note that these seasonal descriptions are not appropriate for all regions, and hence have defined them in the plots by months as well as the text where they are introduced.

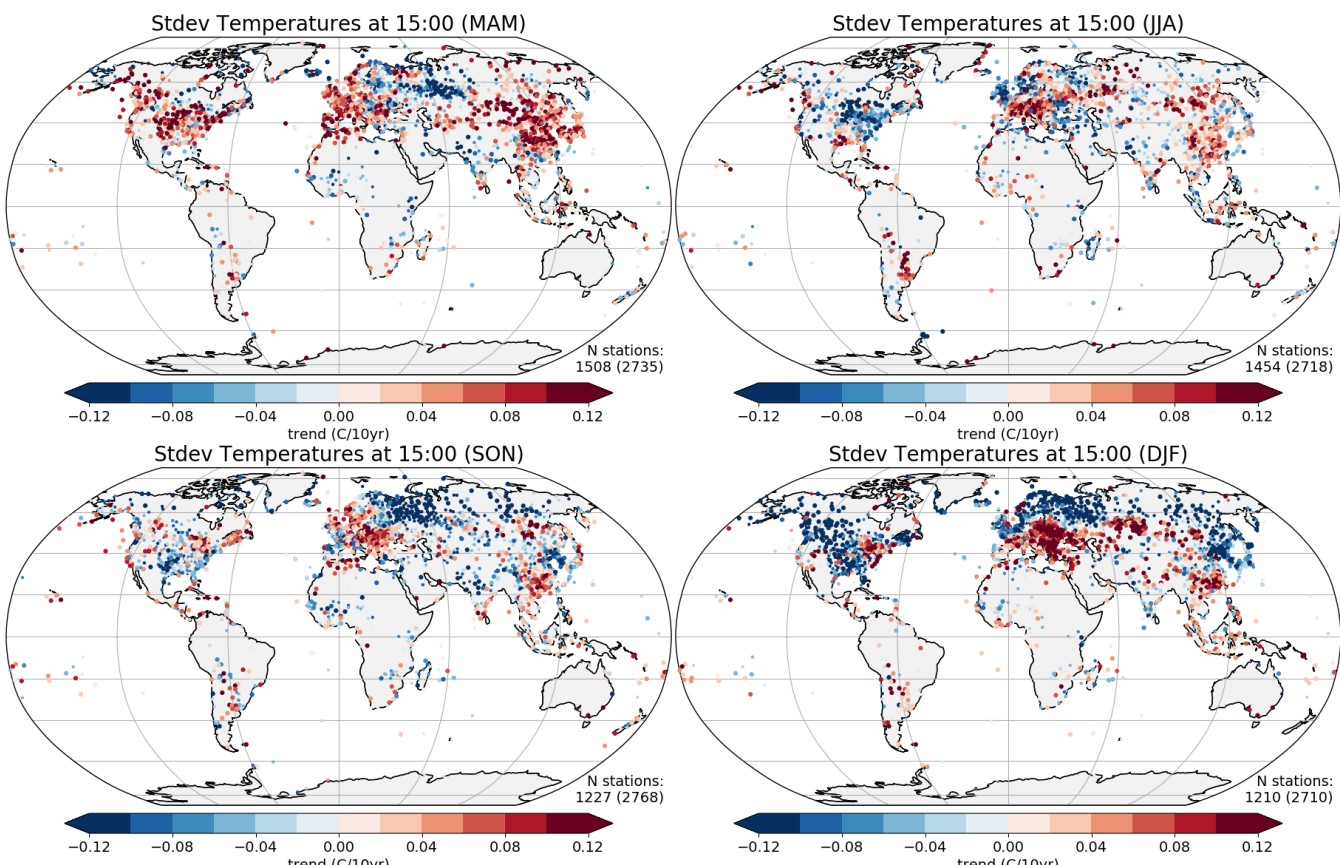

**Figure 8.** Trend over 1974-2018 in distribution standard deviation for temperature at 1500 local time across the four seasons. The stations where the $1\hat{\sigma}$ range of the fitted trend excludes zero are plotted with a larger symbol (the number of these is shown in the bottom right hand corner of each plot). The total number of stations available is shown in parentheses.

temperatures. The HadGHCND dataset uses inverse distance weighting to convert the daily station observations into a gridded dataset. One aspect of this process is that it smooths the data, and also can lead to very few stations contributing to large areas of interpolated values.

Donat and Alexander (2012) find larger regions showing stronger increases in standard deviation around Hudson Bay for both maximum and minimum temperature values than for HadISD. The heterogeneous nature of the changes in the standard

deviation, along with the smaller fraction of stations showing clear trends is in line with the study of Huntingford et al. (2013). For the skew, HadGHCND shows decreasing skew in Greenland, northern Africa, the coasts of eastern Asia and parts of Australia and increases over Europe, in contrast to the results of this study, which in some regions may be the result of low station densities in HadISD (Fig 1). The strong decreases in kurtosis observed close to the tropics in Section 3 are now shown to be concentrated in southern India and south-east Asia, and not always linked to an opposite trend in the standard deviation.

Although McKinnon et al. (2016) link the first four statistical moments derived from GHCND between 1980 and 2015 onto orthogonal basis functions, these still are closely tied to the original statistical moments. Over Europe, the geographical peak of trends in the daytime/maximum mean are both in eastern Europe. However, HadISD does not show such a strong feature of increasing standard deviations across Asia, and GHCND does not show the decreasing daytime standard deviations in northern Europe, but those in eastern Asia do agree. The results from HadISD do not show daytime cooling in the eastern half of North America observed in GHCND. However the general decrease in variance is observed in both. The increase in the central southern region of North America in GHCND is not present here, though the agreement is better by night.

Comparing the seasonal plots with those from Cavanaugh and Shen (2014, 2015) using daily GHCND temperatures over 1950-2010 shows generally good agreement for the changes in the mean. They find the strongest changes over Europe and a clear "warming hole" (Portmann et al., 2009) over the US, especially in JJA and DJF. In HadISD, the strongest changes are over Europe, and the location of the largest values moves between the seasons; but the warming hole is only weakly apparent during JJA over North America. For standard deviation, both Cavanaugh and Shen (2014) and our study find coherent decreases in the Northern Hemisphere, as well as increases in MAM over North America. Over Europe, however, JJA shows the largest differences, with widespread increases in Cavanaugh and Shen (2014, 2015) but changes in both directions are clear in HadISD. McKinnon et al. (2016) also show the decrease in standard deviation over North America, except for a region of increase on the Gulf of Mexico. Also across Eurasia, the patterns are similar in HadISD and McKinnon et al. (2016). The positive trends in skewness found over North America (Cavanaugh and Shen, 2014, 2015) in GHCND are not as prominent in HadISD nor in McKinnon et al. (2016) also using GHCND, with contiguous regions of negative trends in all seasons.

Part of these differences may result from inherent differences in the datasets, for example the differing temporal periods used; HadISD using 1974-2018 compared to GHCND 1980-2015/1950-2010 and HadGHCND 1951-2010. We note that the number of stations reporting in HadISD prior to 1973 is much lower, restricting the possibility of extending our analyses further into the past. Also, here we and McKinnon et al. (2016) do not combine the stations together, unlike in Section 3, Donat and Alexander (2012) and in Cavanaugh and Shen (2014, 2015). The largest difference is that the sub-daily data within HadISD are not the true maximum and minimum values which comprise GHCND and HadGHCND. Although we have used the morning and afternoon values as proxies to the true daily maximum and minimum temperatures, this will not always be the correct match. This does, however, allow the study of how extremes are changing throughout the day. It is difficult to say which of these analyses is the best at determining the change in the distributions of temperature, given the different temporal periods and processing methods, station selections and level of smoothing. Where a number of studies agree on the sign of the change, results are naturally more reliable than regions where they do not.

Changes in the temperature distributions for stations in China are likely to be the result not only of warming on a global scale, but also from more local effects arising from rapid urbanisation (Sun et al., 2016). Impacts from urbanisation are also likely in other regions undergoing rapid development in recent decades. As only a small fraction of the Chinese station network is unaffected by urbanisation (32% in 2015, Wang et al., 2015), any changes are more representative of the urban areas (and hence a larger fraction of the population). Our study period coincides with the rapid urbanisation over China, and as noted above, we see strong increases in the mean of the temperature distributions at all times of the day. Stronger warming is seen

between 06:00 and 12:00 for northern and western parts of China, but not for the southern, coastal regions. Wang et al. (2017) found that the effect of urbanisation was greater on the minimum (morning) temperatures than the maximum (afternoon) ones, which we find in the northern and western parts, but not in the southern, coastal ones (Fig 5). Although China is one example of recent and rapid urbanisation, with resulting effects on the observations as well as people's experience of the weather, other parts of the world will also be affected. This may be to a lesser extent because of a station network better suited to climate monitoring, or the urbanisation occurring over a longer time period.

A number of different causes for changes in temperatures beyond the anthropogenic warming signal, have been suggested. These drivers of change are likely to influence the distributions of observations across the hours of the day in different ways. Large scale circulation changes would change the weather patterns experienced by different regions. Studies across the world have found local and global modes of variability linked to changes in cold and warm extreme temperatures (e.g. Scaife et al. 2008; Kenyon and Hegerl 2008; Barrucand et al. 2008; Sillmann and Croci-Maspoli 2009; Zongxing et al. 2012; Ning and Bradley 2015). Hence changes at these large scale circulation patterns and modes will affect the extremes. However, taking the El Niño Southern Oscillation as an example, there is no consensus on the change under global warming (Stevenson, 2012; Cai et al., 2014; Kim et al., 2014; Vega-Westhoff and Sriver, 2017). That said, as some of the changes in the higher moments shown here have a strong zonal component, especially over Eurasia, this suggests some influence of the poleward expansion of the Hadley Cells (Hu and Fu, 2007) which have been linked to anothropogenic forcings (Lu et al., 2007; Lucas et al., 2014; Tao et al., 2016) on the changes in shape of the temperature distributions.

Other drivers of change in observed temperature distributions have been suggested, including enhanced land-atmosphere coupling and soil-moisture (e.g. Seneviratne et al., 2006; Jaeger and Seneviratne, 2011; Hirschi et al., 2011) or changes in cloudiness and radiation (e.g. Lenderink et al., 2007). In a modelling study of future summer climate over Europe, Fischer and Schär (2008) find that the seasonal cycle as well as interannual and intraseasonal variability have separate effects on the temperature variabiloty, and each of these are affected to a different level by the other drivers.

## 4.2 Dewpoint Temperature Changes

Comparing the changes in the dewpoint temperature moments across the day and the seasonal cycle (Supplementary Information Figs. S17 to S28) with those from the temperatures, shows that on the whole these two variables are quite closely aligned. Despite fewer stations passing the selection criteria for this variable, there are still coherent patterns of change. Over the eight time points during the day, the changes in the mean dewpoints show less variation than the temperatures, most clearly apparent over Europe in the early afternoon. This suggests that the relative humidity over this region has decreased for this part of the day. A similar but less striking decrease in daytime relative humidity is also seen over North America. This decrease in the relative humidity has been observed regionally and globally in the homogenised HadISDH dataset (Willett et al., 2014, 2019), which is based on the HadISD.

The changes in the higher moments of the dewpoint temperature are also reasonably constant through the day, but have a greater spatial heterogeneity and lower magnitude than the temperatures. Notably, there is a clear increase in skew over south-

east Asia during the daytime and early evening, which is not seen in the temperature results, indicating a change in shape away from a low-dewpoint tail towards a high-dewpoint tail.

Seasonally, the greatest change in the dewpoint mean is in SON and DJF over Europe in both the day- and night-time, contrasting with the temperatures of MAM and JJA for the day and also DJF at night. Again the standard deviations and kurtoses are very similar for day and night across all seasons between the dewpoints and temperatures. For the skews, the increase in skew over south-east Asia noted above is most prominent during SON, during the daytime, but is also present at night and differs from the behaviour of the temperatures for all seasons.

### 425   4.3   Wind speed Changes

There are fewer stations with sufficient observations to assess changes in wind speed (Supplementary Information Figs. S29 to S33), and very few in the Southern Hemisphere. As a result we focus our analysis on North America and Eurasia. The proportion of stations where changes in the distribution could be assessed varies between each time step to a greater extent than for the two temperature variables. Despite this, regions with sufficient station densities show a diurnal cycle in the behaviour

of wind speed trends. There are stronger trends for declining wind speeds during the daytime (09:00-18:00) than for the night-time hours. And there are also no large regions that show large increases (at the same magnitude as decreases) in the wind speeds at any time point. However some individual stations do show some increases, for example in south-eastern Europe. For the other three moments (not shown) there are even smaller differences than for the mean for the annual changes over the day.

    We show the seasonal changes at 15:00 for all four moments in Supplementary Information Figs. S30 to S33. For the mean

there are no large changes across the seasons. The changes in standard deviation are also generally unvarying compared to the temperature measures. There are increases in the USA, with stronger values during spring and autumn. All other regions with sufficient station density have decreases in the standard deviation. Both skew and kurtosis show no strong seasonal variations. Skews are increasing over North America but there are no coherent, strong changes elsewhere. The kurtosis is mostly decreasing in south-eastern Europe, and also on the whole across Asia.

### 440   4.4   Discussion of Station-Level Distributions

Across all seasons, times of day and locations, there are strong increases in the means of both temperature and dewpoint temperature with time as expected in a warming world (Stocker, 2014). Although the geographical distribution of stations is not uniform, there appear to be larger increases in Europe, especially during the spring and summer The largest increases in Europe also occur during the daytime rather than night-time, which matches the behaviour seen in Table 1 40-60N (the

445 latitudes corresponding to Europe).The standard deviations show stronger decreases in the northern latitudes, but more regions with increases in the mid- and tropical latitudes in the northern hemisphere. Seasonally, North America shows decreases in the daytime and night-time standard deviation in winter and summer, but increases in spring (Fig. 8 and Supplementary Information Figs. S10 and S14), which was also observed in Cavanaugh and Shen (2015). This indicates a change towards less variability in winter and summer, but a change towards greater variability in spring for both day- and night-time temperatures.

We also note that in Section 3, standard deviations of the temperatures and dewpoint temperatures were decreasing more or less across the whole northern hemisphere, whereas in this study there are regions with clear increases. The zonal study combines the stations together, even if individually any change in the statistical moments over time would not show clear changes ($1\hat{\sigma}$ range of the trends encompassing zero). In contrast, Figs. 3 and 4 emphasise those stations where the $1\hat{\sigma}$ range of fitted trends do not include zero. As noted, this is only around half of the total number of stations and many of the others show
decreasing trends.

Changes in annual skew are different on either side of the North Atlantic, with predominantly increases in North America, and decreases in Europe across the day, and hence a rather small change in the zonally averaged results (Section 3). The seasonal results show increases at higher latitudes (Canada and Scandinavia), and decreases further south (USA and continental Europe) in spring. Other seasons show a more heterogeneous pattern in Europe, but increases across central Asia in summer
and autumn, similar to the patterns from GHCND in Cavanaugh and Shen (2015). In China which was not included in their study, all seasons except winter show decreases in the skews. For the kurtoses, Cavanaugh and Shen (2015) found strong decreases in the southern parts of North America in the summer, which are not seen in HadISD. However, the strong decreases over parts of Europe in winter are present in HadISD.

The observations in HadISD have a finite numerical resolution, with more recent observations being more often recorded
to $0.1^{\circ}$C, and those from earlier parts of the record to $1^{\circ}$C. Underlying this is a more complex process of conversion between Fahrenheit and Celsius (or knots and miles per hour to metres per second), possible truncation and rounding (or vice versa) as well as human induced errors during transcription, digitisation or recording. In the GHCND, Rhines et al. (2015) found that 65% of all temperature observations were misaligned as a result of these effects, and so it is likely that a large fraction of the HadISD observations also have been affected.

Rounding has an effect on the mean and standard deviation dependent on the rounding resolution and any offset in the two scales from zero (rounding width and shift parameter), with zero bias for some combinations (Schneeweiß et al., 2010; Sheppard, 1897), and likely therefore on higher moments as well.

These changes are combined with the discrete measurement times, which also are more likely to be hourly in the more recent period, and more stations will be recording only every three or six hours in the earlier period. More frequent recording results
in a greater likelihood of values close to the true extremes being recorded. Hence, under a stationary climate, this could result in both an increase in the maximum temperatures and a decrease in the minimum temperatures (for a perfect, sinusoidal diurnal cycle). Combined with a warming climate, this could result in an enhancement of the maximum temperatures recorded. There will also have been changes in instruments and recording practices over the period, though the homogeneity assessment should have identified any gross traces these have left in the data record.

It is possible that the effect of these data measurement issues is systematic across the entire global network. We also note that despite the automated quality control and homogeneity assessment of HadISD, there are likely to be data artefacts still present in this global dataset. Therefore further investigation is needed to determine how these two aspects combine to affect the extremes recorded and their change over time in the sub-daily data. However, in our analysis we have looked for contiguous regions of coherent changes for large numbers of stations, spanning geo-political boundaries. Wholesale changes of national

networks and their measurement routines causing systematic biases in the results presented here are likely to stand out as political areas, which are not observed.

## 5 Quantile Regression

Studies of the characteristics of a distribution are useful in describing the overall changes in a meteorological variable. However, they do not make the changes in extremes immediately clear, although they can be calculated through, e.g. extreme value theory.

An alternative way to study the changes in the climate that show changes in past extreme events is to investigate the changes of the quantiles of a distribution themselves. This enables the question of "how much warmer are the warmest 10 per cent of days now than they were in the past?" to be addressed. Quantile regression can be used to calculate trends in specified quantiles of a distribution, rather than trends in the mean (Koenker and Bassett Jr, 1978; Koenker and Hallock, 2001; Barbosa et al., 2011; Franzke, 2015). This allows, for example, cases where there is little or no mean trend but an increased variance over time to

be assessed more completely. In the case of climate observations, and especially given the impact of extreme events on society and infrastructure, quantile regression allows the changes in these extremes over time to be made apparent.

The observations in HadISD naturally contain both an annual and a diurnal cycle. If run purely on these raw data, then the annual upper quantiles are likely dominated by summer daytime temperatures, and the lower by winter night-time temperatures. Although using standard three-month seasons would enable the more detailed study of winter and summer quantiles, those from

500 the shoulder seasons (spring and autumn) would be dominated by the earlier or later parts of these seasons given the strong annual cycle. For this reason McKinnon et al. (2016) use daily maximum and minimum temperature data from GHCND over July and August only to reduce the effect of the seasonal cycle in their study.

Our approach is to use de-seasonlised data, which removes the seasonal aspect described above and also the issue that quantile regression results can be biased when applied to discrete data (Machado and Silva, 2005; McKinnon et al., 2016). The

505 observations in HadISD are at a finite numerical resolution (e.g. temperatures are reported to the nearest $0.1°C$, $0.5°C$ or $1°C$). We use a daily climatology to create hourly anomalies of the HadISD data, retaining the diurnal cycle. Within each calendar day, the numerical resolution remains unchanged by the subtraction of a daily mean from the hourly observations (within-day differences will still have $0.1°C$, $0.5°C$ or $1°C$ resolution). However, etween each calendar day, the different climatology value will result in a different offset, so that between-day differences would not result in discrete values. Hence, overall, the values

passed to the quantile regression algorithm will be less discrete than the raw observations themselves. As discussed in Sections 3.4 and 4.4, we note that the final resolution of HadISD hides the effects of conversion, rounding and truncation (multiple times in various orders) that could occur between measurement and the current use (Rhines et al., 2015).

Daily means were calculated for 1981-2010, requiring at least eight observations in each 24-hour period (equivalent to 3-hourly data). The daily climatology was calculated requiring at least 15 years of data to obtain an average. Finally, the

515 climatology was smoothed using a 5-point binomial filter and subtracted from the sub-daily data for each day to make the climate anomalies. An example of the changes in the percentiles of a timeseries is shown in Fig. 9.

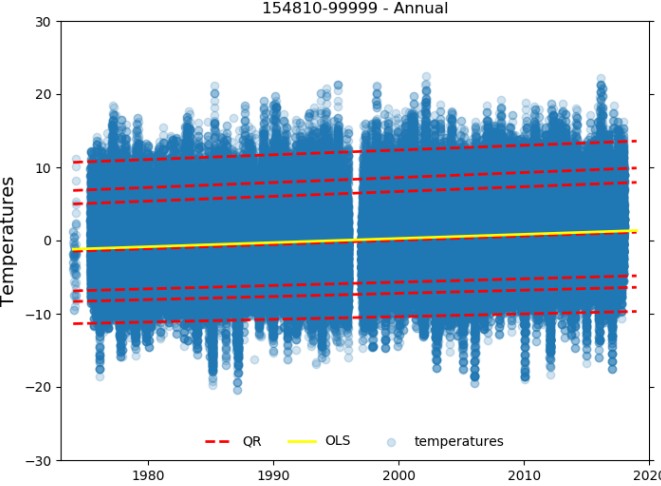

**Figure 9.** An example of the Quantile Regression analysis for a single station, with the trends in the quantiles in red, and the ordinary least squares (OLS) fit in yellow. This is for station 154810-99999, Constanţa Mihail Kogălniceanu International Airport, Romania.

Both Europe and North America have high station densities (see Section 4) compared with other regions, and so our analysis is focussed on these two regions (see also McKinnon et al., 2016). Furthermore, we show the results using a space-filling method, Voronoi tesselation (derived from Delaunay triangulation, Voronoï, 1908; Delaunay, 1934), which uses the perpen-
dicular bisectors of the lines joining each station to split the land surface up into polygons, with each polygon containing a single station. These have then been coloured to show the results of the quantile regression. Stations for which the slope of the quantile shown is not significantly different from zero ($p > 0.05$ as determined by the quantile regression algorithm) are shown in grey, but we do no further significance testing on these trends. It should be noted that areas of high station density will naturally have smaller polygons, but the human eye is drawn to the larger blocks of colour arising from sparse station
networks. However, the advantage of this approach is that the underlying station information is retained, as opposed to it being smoothed when applying some form of gridding method.

We show the changes in the quantiles (0.01, 0.05, 0.1, 0.9, 0.95, and 0.99) over Europe for both JJA and DJF in Figs. 10 and 11 respectively. In some sub-regions large numbers of stations show similar patterns, suggesting that there is a coherent change in the quantile in that area. However, within these there may be isolated stations/cells showing changes in the opposite
direction. Although we have selected those stations that do not have many or large breaks in their timeseries (as determined by PHA, see Section 2), and the HadISD has undergone a number of quality control checks, there will inevitably be a few data quality issues remaining.

The largest change in the summer (JJA) quantiles is observed in south-eastern Europe and western Asia (Fig. 10), focussed on the area north and west of the Black Sea. The increase observed in these regions becomes progressively less strong towards
lower quantiles and also with distance from this region. This indicates that the temperature distribution is broadening over time, with the upper tails changing more rapidly than the lower tails; i.e. the warmest $N$ per cent of events are becoming warmer

as the percentile values change, more rapidly than the cool events (where $N$ is 1, 5, 10 etc). In contrast, the regions close to the North Sea and especially Scandinavia show slight increases for the lower quantiles that becomes less strong for higher quantiles. For the north and west of the UK in general and Scandinavia except western Norway, there are even decreases for the highest quantiles.

Conversely, in the winter (DJF), the strongest increases are in the northern regions and the lower quantiles, with Scandinavia, the Baltic states along with Benelux, Denmark and northern Germany indicating a narrowing of the temperature distribution over time. In the higher quantiles, there are increases for eastern Europe but not as much as in the summer. Westernmost Europe shows very little change across all quantiles. We show the panels for the shoulder seasons (MAM and SON) in the Supplementary Information Figs. S34 and S35. There is a general increase for all the percentiles, and the cold quantiles in the north east of the region shown are warming faster than the rest. But in contrast to Figs. 10 and 11 these increases are much slower and the regions are less coherent. Overall, these results show that warm extremes are becoming more common more quickly during the summer in south-eastern Europe, than elsewhere across the continent. Similarly, cool extremes are becoming less common more quickly during winter in northern Europe than elsewhere.

Franzke (2015) performed quantile regression on daily mean, maximum and minimum temperatures between 1950 and 2013. The mean temperatures show strong regions of increase in both the $5^{th}$ and $95^{th}$ percentiles in eastern Europe and western Russia, especially north of the Black Sea, with the higher percentile region covering a larger area than for the results presented here. When using summer maxima, Franzke (2015) shows a more uniform increase over all of Europe at the higher percentile, except Scandinavia, which behaves similarly in Fig. 10. The magnitudes of the change in the winter temperature percentiles from HadISD are consistent with the studies of Barbosa et al. (2011); Franzke (2015). The behaviour over Turkey stands out in Franzke (2015), but not in HadISD, suggesting a difference in the way these data have been processed or effects arising from the difference in temporal coverage or even station selection.

The difference in behaviour of the lower and upper quantiles indicate a change in the variance of the sub-daily temperatures over Europe, which has already been suggested by the results from Section 4. During the summer, the variance in the large region to the north of the Black Sea, has increased. In contrast, during winter the variance over Scandinavia and northern Europe has decreased. Both of these changes are also clearly visible in Supplementary Information Fig. S10 showing the seasonal change of the standard deviation for each station. Franzke (2015) found a more widespread, uniform decrease in variance in the winter months over Europe, but with a stronger decrease over Scandinavia. However, Franzke (2015) found no increase in variance during the summer in eastern Europe.

Over North America (Supplementary Information Figs. S36 to S39), the trends in quantiles are on the whole smaller than for Europe for most of the year. In the spring (MAM), there are moderate increases in the south-west, which are stronger for the higher quantiles. For the summer (JJA) the higher quantiles show decreases in the northern, central regions, but increases for the lower quantiles in the south west. In the autumn (SON) and winter (DJF) the strongest changes are in the north of the continent at the lower quantiles with more moderate increases in the south and the higher quantiles. Hence the most rapid changes are in the cool extremes during autumn and winter in the north, with less rapid changes in the warm extremes in spring in the south-west. Using GHCND, Rhines et al. (2017) show a similar pattern, with a strong increase in lower percentiles for

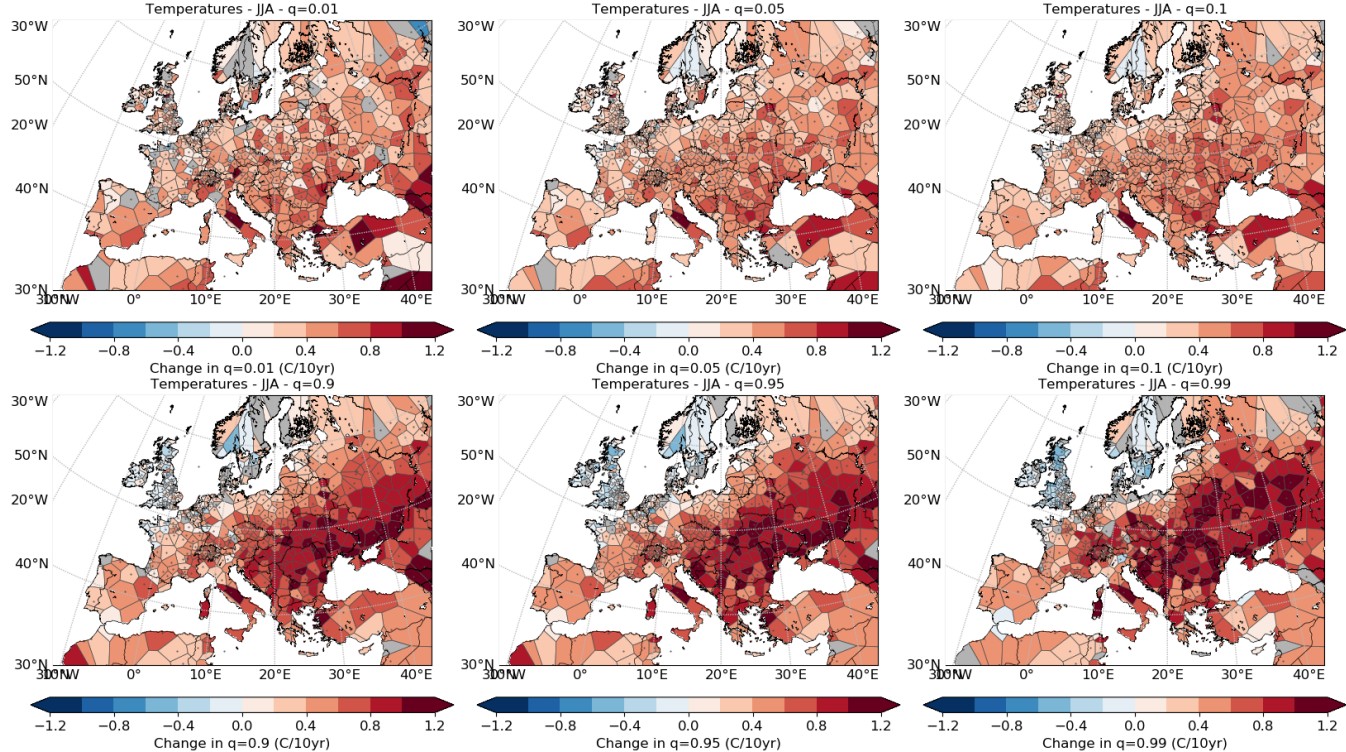

**Figure 10.** Trend over 1973-2017 in quantiles of temperature (°C decade$^{-1}$) at TOP 0.01, 0.05 and 0.10; and BOTTOM 0.90, 0.95, 0.99 over Europe during summer (JJA).

northern latitudes in winter. In the spring their data show decreases in the lower percentiles in the north and increases in the south, almost the opposite in the autumn, and general warming for the minimum temperatures in the summer. We also find the strongest changes for the lower percentiles during winter, indicating a reduction in the range of temperatures during that season.

We also show changes in the dewpoint temperatures (Supplementary Information Figs. S40 to S43) and wind speeds (Fig. S44) over Europe. The dew point temperatures indicate small increases of the quantiles in most regions with no strong regional differences for MAM and JJA for any quantile. However, in SON and DJF, there are stronger increases in the lower quantiles for the north of the region (Scandinavia and north-western Russia), but small increases for other regions and quantiles. So, although the summer warm extremes are becoming more common in south-eastern Europe, there is no corresponding change in the dew point temperatures, suggesting a decrease in the relative humidity. However, in the northern parts, the dew point temperatures are changing roughly in step with the dry-bulb temperatures.

Only the annual changes in wind speeds are shown (Supplementary Information Fig. S44), with some blank tiles and regions because of the smaller number of stations with sufficient observations. There are no large changes in the lower quantiles but clear and larger decreases in the upper quantiles, which matches the behaviour observed in Section 3 and the decrease in

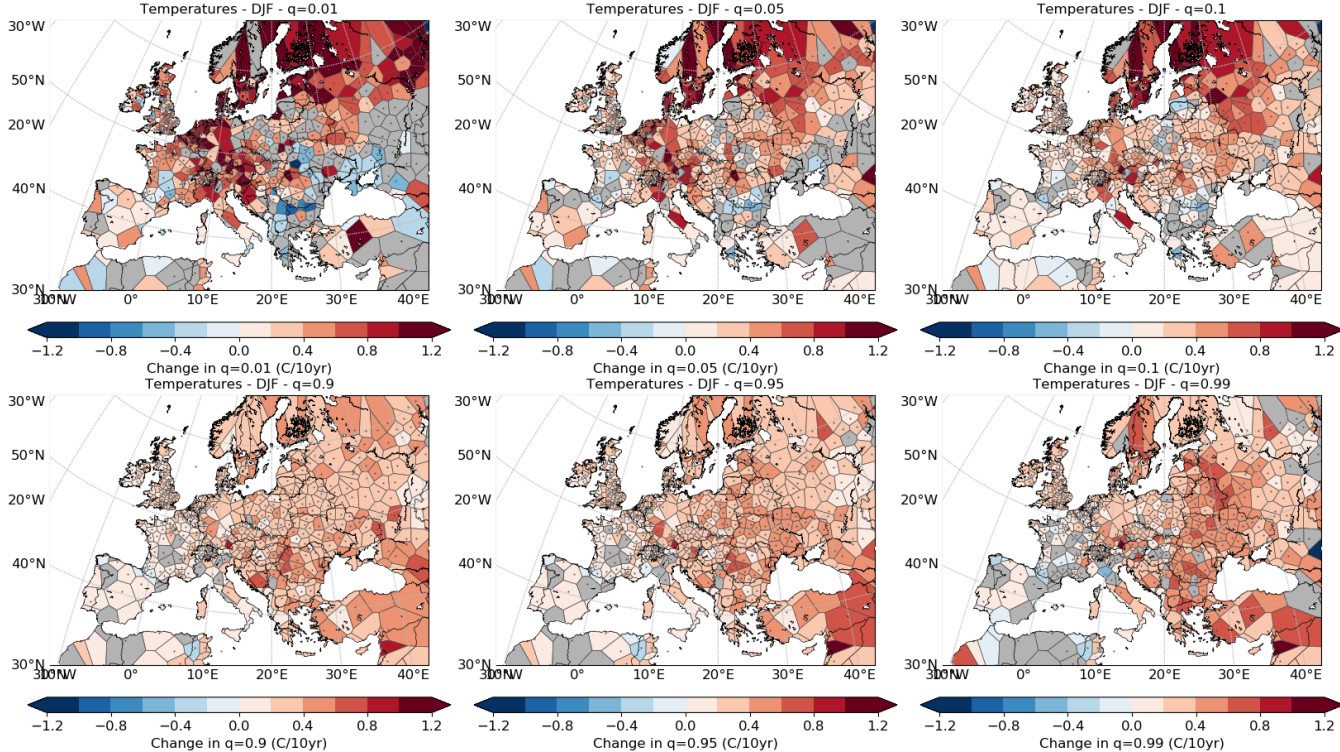

**Figure 11.** Trend over 1973-2017 in quantiles of temperature (°C decade$^{-1}$) at TOP 0.01, 0.05 and 0.10; and BOTTOM 0.90, 0.95, 0.99 over Europe during winter (DJF).

variance observed in Section 4.3. The larger changes in the upper quantiles also suggest the skews are becoming more negative, with decreases in the number and intensities of extremely high wind speeds.

## 6    Summary

To investigate the changes in the shape of the distributions of temperatures and wind speeds in the HadISD dataset, we have
performed three types of assessment. Firstly, by combining the data from the station based observations into zonal averages, the resulting distributions are built from large numbers of observations at the expense of some spatial information. Using five-year bins from 1973 to 2018, we demonstrated the change in the distributions and percentile values. As expected, the mean temperatures increase for both day- and night-time (Stocker, 2014). However, the standard deviations decrease for the northern hemisphere (where there are most stations). Although changes are seen in both tails of the distribution, they are greater in the
lower tail. This behaviour is consistent with both the more rapid change in low temperature extremes (Davy et al., 2017) as well as the decrease in diurnal temperature range (Alexander et al., 2006; Thorne et al., 2016a, b). The dewpoint temperatures show

similar, if more heterogeneous, behaviour in this analysis. Atmospheric moisture, as well as that in the soil may be acting as a brake on the change in the maximum temperatures, resulting in the consistent changes in shape of the distributions observed.

In contrast, the wind speeds showed decreases in the mean, but also in the standard deviations, with larger shifts in the upper tail than the lower. The changes in distributions observed on a zonal level suggest this may be driven by changes in the upper tail. A number of causes for this "stilling" have been proposed, from changes in land-use and circulation patterns to instrumental degradation.

To improve the spatial discrimination, we assessed changes in the distributions for each station individually. The sub-daily nature of HadISD allows these to be calculated across the hours of the local day as well as annually or seasonally. The temperature means show the expected increase over time, but with a daily cycle showing a more rapid increase of mean temperatures during the early afternoon than in the early hours of the morning. However, the higher order moments are much less homogeneous, but have a smaller variation throughout the day. Trends in the standard deviations are on the whole negative, except for a band from the Mediterranean across to central Asia. Conversely, trends in the skew of the temperatures are mainly positive, except over Europe and across into western Asia, indicating increasing warm extremes (positive tail) and decreasing cool extremes. These patterns are in some regions different from those reported in previous investigations by Donat and Alexander (2012) and McKinnon et al. (2016) obtained using (derivatives of) GHCND (true maximum and minimum values). For the seasonal changes, which can be compared to Cavanaugh and Shen (2014), there are also differences to the patterns obtained from GHCND.

These disparities may result from a variety of differences in the datasets and methods. Firstly, the period used in this study is 1974-2018 compared to 1950-2010 (Donat and Alexander, 2012; Cavanaugh and Shen, 2014) and 1980-2015 (McKinnon et al., 2016). Also, by using all of the sub-daily observations in HadISD, this work incorporates the effect of changes across the entire distribution, even if we are concentrating on either the moments or specific percentile values (e.g. all temperature observations rather than just the maxima and minima). The studies using GHCND use the true maximum and minimum values, which allow the focussed investigation of the momentary extremes. We also do not perform any spatial smoothing of the underlying stations, which could result in less variable fields, with smaller extremes.

A number of different drivers for changes in temperatures have been proposed, including changes in global and large scale circulation patterns, land-atmosphere coupling, and cloudiness and radiation. And a number of these have also been shown to be changing as a result of anthropogenic climate change. Over Eurasia, the quasi-zonal patterns of change of the higher moments suggest a link to the expanding Hadley Cells, but further study is required.

Finally we perform a quantile regression analysis to determine how the values of the percentiles have changed, and therefore how the warmest days and coolest nights feel in comparison to others. This approach is in contrast to the ETCCDI indices which use percentiles (e.g. TX90p), which capture the exceedence rate of a fixed percentile value. Here we show that over Europe, the upper percentiles are changing fastest in the large region north-west of the Black Sea in the summer, but the lower percentiles are changing faster in northern Europe and Scandinavia in the winter. Hence the warmest $N$ per cent of summer days are becoming warmer more rapidly than the coolest $N$ per cent in south-eastern Europe, and the reverse for winter days and particularly nights in the north. Coherent changes in dewpoint temperatures are less clear, with increases in the lower

percentiles in autumn and winter for the northern parts of Europe. For the wind speeds, as expected from the other analyses, the higher quantiles show a decline over Europe, with stronger decreases in the east.

All the studies presented herein show widespread increases in mean temperature, and less rapid rises in the dewpoints, as expected from numerous other studies on the observed change in global temperatures. However, the changes in the higher moments are of more interest. Although in many cases (seasons and quantities) the patterns of change are very similar to prior studies (e.g. Donat and Alexander, 2012; Cavanaugh and Shen, 2014, 2015; McKinnon et al., 2016), there are still differences in the spatial patterns between all of the studies. These could arise from differences in the input data (daily extrema versus hourly values) as well as the temporal and spatial coverage. Furthermore, we have used a single, station-based dataset, and differences between datasets (Gross et al., 2018) or the effect of gridding (Cavanaugh and Shen, 2015) can change the apparent behaviour of higher moments. Nevertheless, in these investigations, we have shown that there are changes in the mean, the standard deviations and the skews of the distributions studied. And, when using a sub-daily dataset of temperatures, changes in the number and intensity of extremes appear to arise from the combination of changes in all three of these moments. Further investigations are however needed to determine the underlying drivers for the changes in these distributions, and hence the changes in extremes.

*Data availability.* The HadISD dataset is available under a non-commercial government licence at www.metoffice.gov.uk/hadobs/hadisd

*Author contributions.* Robert Dunn did the majority of the analysis, plotting and writing. Kate Willett and David Parker provided comments and guidance during this work. All authors have contributed text and edits to the main paper.

*Competing interests.* The authors declare there are no competing interests

*Acknowledgements.* We thank Colin Morice for helpful discussions during the course of this work, and Elizabeth Good and Nick Rayner for useful suggestions on the text. This work was supported by the Met Office Hadley Centre Climate Programme funded by BEIS and Defra (GA01101).

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
