# Peer review of "Changes in statistical distributions of sub-daily surface temperatures and wind speed"

_Earth System Dynamics, 2019_

## Referee Comment (RC1) · Anonymous Referee #1 · 26 Aug 2019

Review – "Changes in statistical distributions of sub-daily surface temperatures and wind speed" by Dunn et al.

*General Comments:*

This paper provides a detailed analysis of changes and trends in HadISD temperature and wind data. The analysis uses station data that have been quality controlled and, in some cases, homogenized. Apart from the use of quantile regression to describe changes in the upper and lower tails of the distribution, the paper uses simple statistical methods. Findings are, generally, not surprising, although some trends are found that are somewhat surprising. Comparisons are made with similar papers that have used other datasets.

While this kind of work is fundamental and absolutely necessary, I found the author's apparent decision to limit themselves only to the description of the data unsatisfying. Unfortunately, the paper does not offer physical insight concerning the changes that are observed, and only speculates about the causes of differences between findings reported here and those reported in previous papers based on other datasets. It raises the issue of measurement resolution in the context of quantile regression, but does not concern itself with the impact of measurement resolution on the higher order moments, or how changes in measurement resolution might produce changes in those moments. Also, while the methods are simple, describing them precisely in prose can be very difficult. The authors do write very clearly, but nevertheless, there is sufficient ambiguity in the description of the methods that it would likely not be possible for another scientist to easily reproduce the analyses that are described in the paper. A technical description of the methods using precise mathematical notation could, and should, be included in the supplement.

*Detailed comments:*

Supplement: The contents of the supplement belong to this paper, but the title does not!

40 – There is a bit of confusion here concerning how to refer to extreme value theory. The correct term is "extreme value theory", not "Generalize Extreme Value theory". The GEV (Generalized Extreme Value) distribution emerges from extreme value theory based on one approach to the analysis of extremes – the so-called block maximum approach. Typical practice is to use one-year blocks and to model resulting collection of annual maxima with the GEV distribution. Another approach that is also often used is called the "peaks over threshold" approach, in which exceedances above a high threshold are modelled. This approach leads to the use of the Generalized Pareto (GP) distribution. Both distributions are asymptotic – in the case of the GEV, it is the limiting distribution that is obtained as blocks grow in length without bound (provided there is a limiting distribution). Similarly, the Generalized Pareto distribution is obtained as the level of the threshold increases without bound (again, provided there is a limiting distribution). Thus, it needs to be understood that for a given block length or threshold, the GEV or GP distributions

respectively, can only approximate the distribution of the extremes that are identified. The Brown et al paper you cite uses the peaks-over-threshold approach.

44 – Delete "easily". I think it is debatable whether a full distribution approach is best if the objective is to make inferences about extremes. If this were easy, it is likely that we would not have as much effort as is expended on developing and applying extreme value theory.

56 – "significant" in the statistical sense? Please clarify. To avoid confusion and ambiguity, my suggestion would be to avoid the word "significant" in scientific papers except when discussing statistical significance.

83-85 – To what extent are the moments (particularly the higher moments) sensitive to the degree of inhomogeneity that is permitted? Presumably, consideration of that sensitivity should drive the choice of permitted jump. Otherwise the choice of a number like 1°C, while tidy, would be entirely arbitrary. Note that jumps of this size do not have the same impact on higher-order statistics in all climates; the impact might not be easily discernable in a high variability mid- or high-latitude climate, but it might be very large, in relative terms, in a tropical climate with low temperature variability. This is also where the impact of poor measurement resolution would be the largest.

A second point here is that while temperature is mentioned, nothing is said about winds. Is a similar approach used?

Also, I assume that both dry air temperature and dewpoint temperature are treated in the same way. This raises a small question about whether the same limit for jumps should be used for both, given that the impact on higher order moments may be relatively larger in one case versus the other.

114 – Have you thought of using L-moments rather than ordinary moments? L-moments (linear moments) are generally considered to be somewhat more robust (in the statistical sense of the word), and might be less affected by inhomogeneities.

115 – It is unclear, distribution of what? Is it reasonable, for example, to consider all anomalies (relative to the annual cycle) for the entire year to be part of the same distribution? As an example, the processes that produce variability in summer in the northern mid-latitudes are substantially different from those that do so in winter, so it is not clear, really, what the distribution represents. One might also ask the same question about anomalies that are pooled across latitude bands to produce distributions (should one lump anomalies from different climate types together, and still call it a "distribution"?).

118 – A small quibble here; in statistics, a hat (circumflex) is often added to Greek letters denoting distribution parameters if those parameters have been estimated, as is the case here.

Figure 2 – A small editorial comment is that yellow curves often almost disappear into the background (many people find them hard to see). Another colour scale that doesn't fade away to light colours as you approach the present would be helpful.

161 – What aspect of wind speed does the paper consider? Are these hourly mean values?

163-164 – I think many would dispute that temperature is "normally" (Gaussian) distributed. Indeed, if temperature did have a normal distribution then the study of skewness and kurtosis would be entirely uninteresting, since the normal distribution is fully determined by its first two moments.

167 – Confidence bounds? I'm fine with simply describing +/- one standard deviation as an uncertainty range without a quantified level of confidence, but as soon as you call this a confidence bound, a question arises about the confidence level.

178 – How does seasonality play into these findings about changes in the shape of the temperature distribution? Is there a physical or sampling interpretation to the change in kurtosis that is noted? Could this be due to undetected data artefacts (such as changes in instrumentation over time).

Figure 3 (and others) – the labelling of the figures could be improved throughout the paper and the supplement. In this case, the key piece of information that tells the reader what is in each panel (mean, stdev, etc), is buried in in the middle of a small-font machine generated label.

Figure 3 (and others) – the decision to show only stations with +/- one standard deviation uncertainty ranges that exclude zero seems arbitrary to me. My general preference is not to censor results in this way, since changes in the underlying field are likely relatively smooth and continuous. Imposing a noisy mask (which corresponds to conducting a test with unknown properties) could be argued to be at least as deleterious to describing changes in the dataset as including all of the effects of internal variability and sampling noise at all locations. The locations that are retained are still affected by these uncertainties.

235 – See previous comment.

237-238 – Why is US station density apparently so low?

242 – Replace "Over 90%" with "Approximately 95%" since 2806/2956=0.949.

257 – "visible in more widely in" → "visible more widely in"

272 – I don't see how greater station density would increase trends (do stations emit heat? ☺).

286-295 – Here and elsewhere, it would be really useful to have more depth in the analysis of differences between datasets (or analyses). There is merit in pointing out differences, of course, but it would be much more useful to those assessing datasets if the authors could delve into the causes of these differences.

297-303 – What fraction of stations are affected by expanding urban heat islands, which can induce apparent trends when formerly rural stations come to be affected by urban heat islands as nearby cities develop? This is a huge problem in China, for example. China has a very dense observing network, but it was not built for climate purposes, and only a small fraction of Chinese stations can be classified as being rural throughout their observing history (this is particularly the case in Eastern China). It has been estimated that this difficult-to-detect cause of inhomogeneity has resulted in recorded temperature in China warming substantially more than the country has actually warmed (despite intense development, urban areas still represent only a small fraction of the Chinese land area). See Sun et al., 2016, *Nature Climate Change.* Other parts of the developing world are, presumably, similarly affected.

328-330 – It would be really useful if the authors could dig more deeply and provide more than speculation on the causes of the differences that are noted.

337-338 – Is this decrease in relatively humidity confirmed in HadISDH?

392 – See a previous comment pointing out that changes in the tails are not necessarily easily inferred from changes in the full distribution.

395 – Replace "calculates" with "can be used to calculate". Implicitly, the statement here describes quantile regression as calculating linear trends. The method, is in fact, a lot more flexible than that (other trend models can also be fitted, if appropriate).

408 – As mentioned previously, I think the question of measurement (or perhaps better, data recording) resolution should have been introduced much sooner. Also, the statement that temperatures are reported to the nearest 0.1°C or 1°C does not really convey the full complexity of the problem (e.g., associated with conversion of °F to °C, and the subsequent rounding or truncation to increments of 0.1°C or 1°C). See Rhines et al, 2015.

421 – How many stations in a polygon?

435 – Define N.

451 – This sentence seems to be grammatically challenged.

500-501 – Have there been changes in instrumental design or the processing of instrumental readings (e.g., approaches that might have been applied to compensate for variations in cup anemometer drag with velocity) that might have contributed to the apparent "stilling"?

519 – I'm not sure that spatial smoothing would result in "less extreme fields". The idea that the magnitude of extremes could be reduced is reasonable, but the idea that the result could be "less extreme" seems less well founded. "Extremeness", the relative position of an observation in the tail of its distribution, can only be evaluated within the context of the variability that a given data product tries to represent. Saying that the magnitudes of point rainfall observations can be larger than grid box mean rainfall amounts doesn't help one decide whether a given grid box mean value is more "extreme" (situated deeper in the tail) than a given point observation.

---

## Referee Comment (RC2) · Anonymous Referee #2 · 10 Sep 2019

This manuscript studied the changes in statistical distributions of sub-daily surface temperatures, dewpoint temperatures, as well as wind speeds, using station-based HadISD dataset. Both zonally averaged quantities and the spatial distributions were considered, and a quantile regression analysis was also performed. Besides the changes of the mean values, different statistical moments were also studied. This work provided great details about the changes of the temperatures and wind speed, in context of global warming. Roughly speaking, I think this manuscript can be a good reference for people who studies the effects of global warming. However, to publish this work in ESD, there are several issues that need to be addressed.

1, It is difficult to catch the highlights of this work. Many calculations have been done in this work, but by reading the manuscript, it is very easy to get lost. I would suggest

the authors to make a better discussion, and the conclusion should be improved.

2, There are many figures in the supplementary materials. But the main text discussed these figures frequently. It seems that the figures are important. Therefore, why not include these figures in the main text? Or maybe the structure of the manuscript needs to be improved. Moreover, for the figures in the supplementary materials, I would suggest the authors use "Fig. S1, S2, etc.", to distinguish from the figures in the main text.

3, When studying the changes, what is the statistical significance level? What method was used to do the significance test? Why use $1\sigma$ as the threshold?

4, Since only data over the past 45 years were analyzed. Are the observed changes influenced by potential decadal variabilities in the climate system? Can the statistical significance test rule out the potential influences from the decadal variabilities?

5, The results from this work were compared with the findings from previous studies. When the results are not in line with each other, which results are more reliable? Why? The authors may need to better explain why the results are different.
* * *

---

## Author Comment (AC1) · 2 Oct 2019

Review – "Changes in statistical distributions of sub-daily surface temperatures and wind speed" by Dunn et al.

General Comments:

This paper provides a detailed analysis of changes and trends in HadISD temperature and wind data. The analysis uses station data that have been quality controlled and, in some cases, homogenized. Apart from the use of quantile regression to describe changes in the upper and lower tails of the distribution, the paper uses simple statistical methods. Findings are, generally, not surprising, although some trends are found that are somewhat surprising. Comparisons are made with similar papers that have used other datasets.

Thank you. We have addressed your comments individually below, and marked in the revised manuscript where we have made changes, except for updated tables and figures, or if paragraphs have been moved into a new section.

While this kind of work is fundamental and absolutely necessary, I found the author's apparent decision to limit themselves only to the description of the data unsatisfying. Unfortunately, the paper does not offer physical insight concerning the changes that are observed, and only speculates about the causes of differences between findings reported here and those reported in previous papers based on other datasets.

This is partially because of the expertise of the authors and our desire as outlined in the introduction to find out what the changes have been, rather than assess these changes against possible explanations. We have now mentioned a number of analyses which have suggested causes for changes in the distribution of temperatures, and hence also the extremes.

It raises the issue of measurement resolution in the context of quantile regression, but does not concern itself with the impact of measurement resolution on the higher order moments, or how changes in measurement resolution might produce changes in those moments.

Given the blending across stations in the zonal analysis we do not address this point in detail there, but do raise it in the summary of that section. A more detailed discussion is now given in the summary of Section 4, which addresses both the measurement resolution and also the discrete temporal nature of HadISD, which also changes over time. We note that there will be an effect from these but are not able to quantify it with the analysis we have performed here. However, given changes in measurement are likely to occur on a national basis over time, we note that individual geo-political regions do not stand out (which could indicate a bias resulting from measurement or processing practices).

Also, while the methods are simple, describing them precisely in prose can be very difficult. The authors do write very clearly, but nevertheless, there is sufficient ambiguity in the description of the methods that it would likely not be possible for another scientist to easily reproduce the analyses that are described in the paper. A technical description of the methods using precise mathematical notation could, and should, be included in the supplement.

We have added a more technical description on the data pre-processing and how the selection criteria apply at the beginning of the supplementary information, which we hope clarifies how we have prepared the data for our analysis. The trend calculation (median of pairwise slopes) and quantile regression are standard algorithms, and so we have not reproduced these.

Detailed comments:

Supplement: The contents of the supplement belong to this paper, but the title does not!

Thank you - corrected

40 – There is a bit of confusion here concerning how to refer to extreme value theory. The correct term is "extreme value theory", not "Generalize Extreme Value theory". The GEV (Generalized Extreme Value) distribution emerges from extreme value theory based on one approach to the analysis of extremes – the so-called block maximum approach. Typical practice is to use one-year blocks and to model resulting collection of annual maxima with the GEV distribution. Another approach that is also often used is called the "peaks over threshold" approach, in which exceedances above a high threshold are modelled. This approach leads to the use of the Generalized Pareto (GP) distribution. Both distributions are asymptotic – in the case of the GEV, it is the limiting distribution that is obtained as blocks grow in length without bound (provided there is a limiting distribution). Similarly, the Generalized Pareto distribution is obtained as the level of the threshold increases without bound (again, provided there is a limiting distribution). Thus, it needs to be understood that for a given block length or threshold, the GEV or GP distributions respectively, can only approximate the distribution of the extremes that are identified. The Brown et al paper you cite uses the peaks-over-threshold approach.

Thank you for this clarification - we have amended and expanded this section in light of your comments to read: "The "block maximum" approach
models a set of e.g. annual, maxima with a Generalised Extreme
Value distribution (e.g. Christidis et al, 2011). Alternatively, the "peaks over threshold"
approach models all peaks over a fixed threshold with the Generalised
Pareto distribution (e.g. Brown et al 2008)"

44 – Delete "easily". I think it is debatable whether a full distribution approach is best if the objective is to make inferences about extremes. If this were easy, it is likely that we would not have as much effort as is expended on developing and applying extreme value theory.

Done

56 – "significant" in the statistical sense? Please clarify. To avoid confusion and ambiguity, my suggestion would be to avoid the word "significant" in scientific papers except when discussing statistical significance.

In this case we are reporting the results in Donat et al, 2012, but we have added the significance level from their discussion. The only other mention of significance is in the section on Quantile Regression, where a p=0.05 value is used and stated.

83-85 – To what extent are the moments (particularly the higher moments) sensitive to the degree of inhomogeneity that is permitted? Presumably, consideration of that sensitivity should drive the choice of permitted jump. Otherwise the choice of a number like 1°C, while tidy, would be entirely arbitrary. Note that jumps of this size do not have the same impact on higher-order statistics in all climates; the impact might not be easily discernable in a high variability mid- or high-latitude climate, but it might be very large, in relative terms, in a tropical climate with low temperature variability. This is also where the impact of poor measurement resolution would be the largest.

We have added the following sentence to highlight the limitation of the PHA to find jumps for station-variable combinations with high variability: "This used monthly averages derived from the sub-daily HadISD observations to compare station-pair differences to identify potential jumps in timeseries, a process which is more effective for stations and variables with low variability."

At one level, the choice of 1C is arbitrary and tidy. However there are reasons behind the choice. In the homogeneity assessment of HadISD, the distribution of all inhomogeneities is shown in Dunn et al, 2014, Figure 2. The majority of all inhomogeneities are smaller than 1C, and our intention by excluding stations with a jump of more than 1C is to remove those with the greatest inhomogeneities (the most contaminated stations). Around 15% of stations have no breaks in temperature and around 7% for wind speed (Dunn et al, 2016 - HadISD v2), and the average inhomogeneity is around 0.8-0.9C or m/s (the largest of both methods for each variable – average and diurnal temperature range for the temperatures, and average and maximum for the wind speed). In an example assessment shown by Dunn et al (2014) the effect of reducing the number of stations available for an analysis by being more and more restrictive on the maximum inhomogeneity allowed showed that the smallest rms difference between monthly means in HadISD and CRUTEM was reached for a maximum inhomogeneity between 1 and 2C. The result at 0.5C indicated that the resulting small station network limited the analysis. This is what we are attempting to balance in this analysis, a reasonable station network resulting from the exclusion of the most contaminated stations. We now refer explicitly to this assessment at this point in the text. We have also corrected the maximum number of inhomogeneities allowed as well as the selection criteria which used the total number since 1931 (start of HadISD record), but now uses 1974 (and 1973 for the quantile regression section).

The effects of heterogeneity on the higher moments of sub-daily data are unfortunately complex, being . dependent on the weather at the station at each time point, not just the value of the single variable being assessed. Hence, from a monthly inhomogeneity alone, it is in our minds difficult to say how the moments of distributions will be affected from a physical perspective. However, reverting to the distribution of the inhomogeneities in Dunn et al, (2014), the distribution of the estimate jumps (rather than the fitted curve) shows no clear asymmetry for the central portions. The means of both the estimated and fitted distributions are not exactly zero, but correspond to fatter tails on one side or the other, the stations contributing to which are excluded in our analysis as they have inhomogeneities >1C or >1m/s. Therefore any systematic effect of the remaining inhomogeneities can be argued to be small, given this analysis is looking at combinations or larger regions of stations rather than small-numbers where single jumps could cause issues. We have added the following paragraph at this point in the text:
As the selected stations still contain uncorrected inhomogeneities, we note that these will have affected the analysis of the statistical properties of the observations outlined herein. But the distributions of the estimated inhomogeneities as shown in Dunn et al. (2014, 2016) do not have a strongly non-zero mean, and the central portions retained by the approach above have no clear asymmetry. Therefore, as the following analyses use combinations of stations or look for contiguous regions of change that cross national (i.e. observing-practice) boundaries, we do not expect large effects arising from the remaining inhomogeneities themselves.

A second point here is that while temperature is mentioned, nothing is said about winds. Is a similar approach used?

Yes, a similar approach, using 1m/s. We have now noted this at this point.

Also, I assume that both dry air temperature and dewpoint temperature are treated in the same way. This raises a small question about whether the same limit for jumps should be

used for both, given that the impact on higher order moments may be relatively larger in one case versus the other.

Yes, they are treated the same way, with the same limit. We have clarified this in the text. The distribution of dewpoint inhomogeneities is in fact a little wider than the temperature one from Dunn et al, 2014, and so this threshold is more restrictive in this case, and hence a smaller network.

114 – Have you thought of using L-moments rather than ordinary moments? L-moments (linear moments) are generally considered to be somewhat more robust (in the statistical sense of the word), and might be less affected by inhomogeneities.

We had not come across L-moments in our reading, so thank you for bringing these to our attention. We had noted the use of Legendre polynomials as basis functions in McKinnon et al (2016) and so were aware that other methods of calculating the changes in shape of a distribution exist (we note this in the text). They are something that we will investigate for future analyses of this type. However, for this study our goals were to be able to easily compare with and build on previous work using the ordinary moments, as used in Donat et al (2012), Cavanaugh & Shen (2014), which we think are also easier for readers without deeper statistical knowledge to understand. We have added the following paragraph at this point in the text to draw attention to the fact that other moments exist, and our reasons for sticking with the ordinary set:

Other statistical moments, e.g. linear (or L-) moments (Hosking, 1990) have also been used for the analysis of changes in the characteristics of distributions (e.g. Fowler et al, 2005, Simolo et al, 2011). However, as noted above, we use the ordinary statistical moments to enable clearer comparison with previous, similar analyses (e.g. Donat et al, 2012).

115 – It is unclear, distribution of what? Is it reasonable, for example, to consider all anomalies (relative to the annual cycle) for the entire year to be part of the same distribution? As an example, the processes that produce variability in summer in the northern mid-latitudes are substantially different from those that do so in winter, so it is not clear, really, what the distribution represents. One might also ask the same question about anomalies that are pooled across latitude bands to produce distributions (should one lump anomalies from different climate types together, and still call it a "distribution"?).

As outlined at the beginning of this section, our aim was to expand upon the study from Donat et al, (2012), who used only 3 zonal bands to compare between two 30-year periods. In this section of the study, we do de-seasonalise the data using a climatology, so that the tails of the distribution are not dominated by e.g. summer and winter for the temperatures in the mid-to-high latitudes. But we agree that the underlying processes are going to be different across the year, and across different longitudes within a zonal band as well. We have added "on an annual basis" at the end of the first sentence of this section, and "at the expense of spatial resolution" at the end of the second sentence.

The limitations of the zonally averaged analysis are addressed in the summary of this section (3.4), which then leads on to the following analysis (section 4) where observations from each station are assessed individually, seasonally and at different times throughout the day. We have expanded the final sentence of 3.4 (was line 199-201) to say the following:
"This analysis combines station anomalies together in zonal bands, separately for local day- and night-time. In doing so, it removes any small scale, regional changes in the characteristics of the distributions of the meteorological variables. We have also not split the analysis up into seasons (neither 3-month or wet/dry), and so large changes occurring in only part of the year will have been

diluted in this analysis.  In the following section we improve the assessment of regional and seasonal changes but in doing so reduce the number of observations available to characterise the distributions."

We have added "of these zonally averaged quantities" at the beginning of the 4th paragraph of Section 3 for clarity.

118 – A small quibble here; in statistics, a hat (circumflex) is often added to Greek letters denoting distribution parameters if those parameters have been estimated, as is the case here.

Added

Figure 2 – A small editorial comment is that yellow curves often almost disappear into the background (many people find them hard to see). Another colour scale that doesn't fade away to light colours as you approach the present would be helpful.

This figure has used the "Viridis" colourmap which has been designed for readers who are colourblind, is perceptively uniform and should also work in black and white. Most sequential colourmaps which work well for those with colourblindness use intensity as well as hue, and one end is most often rather light.  Of those available as standard in Python, Viridis is one of the darkest at the "light" end (https://matplotlib.org/tutorials/colors/colormaps.html).  Other (diverging or qualitative) colourmaps are not appropriate given these plots show a sequence of 5-year values.  However, we have increased the line width in the updated plots to assist the clarity.

161 – What aspect of wind speed does the paper consider? Are these hourly mean values?

The values in the HadISD are for example 1 or 2 minute average values over the US, as noted by DeGaetano, (1997) , or 10 minute averages in the UK. in relation to parent datasets to the ISD. We've added to this sentence and include the reference.

163-164 – I think many would dispute that temperature is "normally" (Gaussian) distributed. Indeed, if temperature did have a normal distribution then the study of skewness and kurtosis would be entirely uninteresting, since the normal distribution is fully determined by its first two moments.

Thank you - we have removed this sentence.

167 – Confidence bounds? I'm fine with simply describing +/- one standard deviation as an uncertainty range without a quantified level of confidence, but as soon as you call this a confidence bound, a question arises about the confidence level.

We have replaced this with "an uncertainty range"

178 – How does seasonality play into these findings about changes in the shape of the temperature distribution? Is there a physical or sampling interpretation to the change in kurtosis that is noted? Could this be due to undetected data artefacts (such as changes in instrumentation over time).

As noted in your comment regarding line 115, the analysis in this section is done only for annual data, rather than being split into seasons, and we have now added a mention and discussion of this

to the beginning and end of this section respectively. Also, a climatology has been subtracted from the observations to remove the impact of seasonal effects on these results.

Given the small number of stations in the tropical latitude bands, and hence fewer observations to constrain the distribution at in each 5-year interval, we do not want to speculate on possible causes of the kurtosis change, also given the high-order of this moment. We have referred to these changes in kurtosis in Section 4, as these negative changes are concentrated in south-eastern Asia, but have no definitive explanation. As now noted further on in the manuscript, changes in circulation patterns and modes of variability, cloudiness and land-atmosphere coupling could all affect these distributions. More detailed analysis in a regionally focussed paper would be needed to determine the likeliest underlying causes.

Despite the quality control and homogeneity assessment of the HadISD, there will still be undetected data artefacts present which, for small station numbers may have a greater impact. We raise this along with the discussion on resolution.

Figure 3 (and others) – the labelling of the figures could be improved throughout the paper and the supplement. In this case, the key piece of information that tells the reader what is in each panel (mean, stdev, etc), is buried in in the middle of a small-font machine generated label.

We have increased the fontsize, improved the titles and moved some of the other text to help clarify what is shown on these scatter plots.

Figure 3 (and others) – the decision to show only stations with +/- one standard deviation uncertainty ranges that exclude zero seems arbitrary to me. My general preference is not to censor results in this way, since changes in the underlying field are likely relatively smooth and continuous. Imposing a noisy mask (which corresponds to conducting a test with unknown properties) could be argued to be at least as deleterious to describing changes in the dataset as including all of the effects of internal variability and sampling noise at all locations. The locations that are retained are still affected by these uncertainties.

Earlier versions of this analysis did include all the stations, with larger plotted symbols for those where the +/- 1 standard deviation excluded zero. However the result of discussions in the author team and wider, those stations where the +/- range included zero were removed, resulting in the current figures, our thinking at the time being that the trends were less reliable even if they were large in magnitude.

In light of your comments our previous discussions, we have reverted to showing all the stations on these plots. However, we do emphasise those where the range excludes zero as there is greater confidence for the trends in these stations to not be artefacts resulting from (a) the relatively small number of temporal bins being fitted, and (b) the use of a linear model for the change over time.

As a result, all the figures for Section 4 have been updated, along with those in the supplement.

235 – See previous comment.

This has been updated in light of the new figures prepared in response to the previous comment

237-238 – Why is US station density apparently so low?

This is the result of the station selection driven from the homogeneity assessment. As erroneously (and now corrected) we had used the number of inhomogeneities since the start of the HadISD record (1931) rather than 1974 and many of the US stations have long records, these were being excluded. This has been corrected, thank you for bringing this to our attention. As a result, all figures for Section 3 and 4 as well as the tables for Section 3 have been updated.

242 – Replace "Over 90%" with "Approximately 95%" since 2806/2956=0.949.

Done, with updated numbers in light of the correction from the previous comment

257 – "visible in more widely in" à "visible more widely in"

Done

272 – I don't see how greater station density would increase trends (do stations emit heat?).

We have rephrased this sentence as follows "The region with the largest trends is Europe. However, it also has the highest station densities, and so the eye is more drawn to this region than areas which also show large trends but from a sparser station network."

286-295 – Here and elsewhere, it would be really useful to have more depth in the analysis of differences between datasets (or analyses). There is merit in pointing out differences, of course, but it would be much more useful to those assessing datasets if the authors could delve into the causes of these differences.

In light of comments from reviewer 2, we have re-ordered the paragraphs in the temperature part of Section 4, adding in a discussion sub-heading. We have there expanded the paragraph which outlines the differences between this and the other studies. As well as having its own temporal coverage, station selection, and processing, HadISD is sub-daily and so can show the changes over the course of the day rather than just for the (un-timed) point extremal values.

297-303 – What fraction of stations are affected by expanding urban heat islands, which can induce apparent trends when formerly rural stations come to be affected by urban heat islands as nearby cities develop? This is a huge problem in China, for example. China has a very dense observing network, but it was not built for climate purposes, and only a small fraction of Chinese stations can be classified as being rural throughout their observing history (this is particularly the case in Eastern China). It has been estimated that this difficult-to-detect cause of inhomogeneity has resulted in recorded temperature in China warming substantially more than the country has actually warmed (despite intense development, urban areas still represent only a small fraction of the Chinese land area). See Sun et al., 2016, Nature Climate Change. Other parts of the developing world are, presumably, similarly affected.

Thank you for raising this point. As noted in our introduction, we are interested in the experience of people, environment and infrastructure of extreme events and how these have changed, so the urbanisation effect is part of this. A discussion of this and how it can influence the results is relevant to this work, and we have added a paragraph to the end of section 4.1.1.

328-330 – It would be really useful if the authors could dig more deeply and provide more than speculation on the causes of the differences that are noted.

This paragraph has been expanded in the new subsection discussing these temperature results and also includes the important difference that HadISD is a sub-daily dataset, rather than just a max/min. Hence the using afternoon/morning temperatures as proxies for max/min will not always be appropriate.

337-338 – Is this decrease in relative humidity confirmed in HadISDH?

It is (see Willett et al, 2014, Willett et al, 2019), but as HadISDH is based on HadISD (but undergoes extra homogenisation and gridding) this is comparatively unsurprising. We have added "This decrease in the relative humidity has been observed regionally and globally in the homogenised HadISDH dataset (Willett et al, 2014, Willett et al, 2019), which is based on the HadISD." at this point for completeness.

392 – See a previous comment pointing out that changes in the tails are not necessarily easily inferred from changes in the full distribution.

We have replaced "inferred" with "calculated through, e.g. extreme value theory" to link back to the discussion in the introduction.

395 – Replace "calculates" with "can be used to calculate". Implicitly, the statement here describes quantile regression as calculating linear trends. The method, is in fact, a lot more flexible than that (other trend models can also be fitted, if appropriate).

Done

408 – As mentioned previously, I think the question of measurement (or perhaps better, data recording) resolution should have been introduced much sooner. Also, the statement that temperatures are reported to the nearest 0.1°C or 1°C does not really convey the full complexity of the problem (e.g., associated with conversion of °F to °C, and the subsequent rounding or truncation to increments of 0.1°C or 1°C). See Rhines et al, 2015.

We agree that there are deeper issues than just the apparent reporting to the nearest 0.1, 0.5 or 1 degree. However, we feel that the quantile regression section is not the right place to raise these issues, as the effects of using discrete data are separate to the effect that could occur in the quantile regression algorithm if de-seasonalisation wasn't done. As noted above, we have added extra sections to raise this issue elsewhere in the manuscript.

421 – How many stations in a polygon?

One. We have added a clause to highlight this

435 – Define N.

In this case N is meant to represent whatever upper percentile the reader wishes to insert (1, 5, 10, etc) rather than a denoting a specific quantity. For clarity we have added "where N=1, 5, 10 etc"

451 – This sentence seems to be grammatically challenged.

Thank you. The Franzke 2015 reference at the beginning has been removed as this seems to have been an erroneous paste. It now reads "The mean temperatures show strong regions of increase in both the 5th and 95th percentiles in eastern Europe and western Russia, especially north of the

Black Sea, with the higher percentile region covering a larger area than for the results presented here"

500-501 – Have there been changes in instrumental design or the processing of instrumental readings (e.g., approaches that might have been applied to compensate for variations in cup anemometer drag with velocity) that might have contributed to the apparent "stilling"?

We have added a sentence at this point outlining a number of the possible explanations that have been put forward for the stilling, including issues with wind sensors. Azorin-Molina et al, (2018) show that older instruments measure lower wind speeds than new ones in a comparative field trial but more investigations are needed before being able to confirm whether this is a (partial) explanation for the stilling.

519 – I'm not sure that spatial smoothing would result in "less extreme fields". The idea that the magnitude of extremes could be reduced is reasonable, but the idea that the result could be "less extreme" seems less well founded. "Extremeness", the relative position of an observation in the tail of its distribution, can only be evaluated within the context of the variability that a given data product tries to represent. Saying that the magnitudes of point rainfall observations can be larger than grid box mean rainfall amounts doesn't help one decide whether a given grid box mean value is more "extreme" (situated deeper in the tail) than a given point observation.

Thank you for pointing out this interpretation of our sentence. We had intended to convey that the magnitude of the extremes could be reduced by spatial smoothing compared to an unsmoothed representation, and agree that the relative "extremeness" of the tails of the smoothed fields is not affected. We have rephrased this as "which could result in less variable fields, with smaller extremes"

---

## Author Comment (AC2) · 2 Oct 2019

This manuscript studied the changes in statistical distributions of sub-daily surface temperatures, dewpoint temperatures, as well as wind speeds, using station-based HadISD dataset. Both zonally averaged quantities and the spatial distributions were considered, and a quantile regression analysis was also performed. Besides the changes of the mean values, different statistical moments were also studied. This work provided great details about the changes of the temperatures and wind speed, in context of global warming. Roughly speaking, I think this manuscript can be a good reference for people who studies the effects of global warming. However, to publish this work in ESD, there are several issues that need to be addressed.

Thank you. We have addressed your issues individually below, and marked in the revised manuscript where we have made changes, except for updated tables and figures, or if paragraphs have been moved into a new section.

1, It is difficult to catch the highlights of this work. Many calculations have been done in this work, but by reading the manuscript, it is very easy to get lost. I would suggest the authors to make a better discussion, and the conclusion should be improved.

In light of comments by the other review we have added some extra paragraphs in the discussion/summary sections for each analysis. Given the length of the sections addressing temperature for the station plots, we have reordered these paragraphs, adding in an extra discussion on the temperatures, including references possible causes for the changes observed, before moving on to the other two variables. The summary of Section 4 now also includes aspects of the observational data and how these could have affected the results. Hopefully by including some of the figures from the supplement, as you suggest below, this also helps readers.

We have made some changes to the final summary to highlight some of the other changes further up in the manuscript, and tried to clarify this section.

2, There are many figures in the supplementary materials. But the main text discussed these figures frequently. It seems that the figures are important. Therefore, why not include these figures in the main text? Or maybe the structure of the manuscript needs to be improved. Moreover, for the figures in the supplementary materials, I would suggest the authors use "Fig. S1, S2, etc.", to distinguish from the figures in the main text.

We were attempting to strike a balance of the number of figures in the manuscript, and not have too many to dominate the text. As we discuss up to three variables, annually and in some cases seasonally, and with a sub-daily dataset are able to split across the day as well for up to 4 moments too, we didn't want to overload the manuscript with figures.

In light of the above comment we have added a number from the supplement into the main body of the paper, but have also retained them in the supplement so that this still has a logical flow to assist readers. This increases the duplication, but we feel this is appropriate to assist readers using the supplement.

We have also updated the numbering as suggested.

3, When studying the changes, what is the statistical significance level? What method was used to do the significance test? Why use $1\sigma$ as the threshold?

In this analysis, except for the quantile regression section, we do not assess the statistical significance level of any changes. We had used the +/-1σ range to determine how reliable a trend is by whether this range includes or excludes zero, and only plot those on the maps. In light of the comments by Referee 1, and as noted in our response to their comments about Figure 3 we have reverted to showing all stations in these scatter plots. We emphasise those where the +/-1σ range excludes zero with a larger symbol as the trends are more likely to be reliable but all the stations are now plotted.

Our aim with this approach is to balance occasions where there is a large trend magnitude but also a large spread in the possible trend values from the median of pairwise slopes algorithm. There are many studies which discuss the advantages and disadvantages of any form of significance testing, and the care required to frame both the problem and the test in the correct way (e.g. Ambaum, M.H., 2010: Significance Tests in Climate Science. J. Climate, 23, 5927–5932, https://doi.org/10.1175/2010JCLI3746.1, Wilks, D.S., 2016: "The Stippling Shows Statistically Significant Grid Points": How Research Results are Routinely Overstated and Overinterpreted, and What to Do about It. Bull. Amer. Meteor. Soc., 97, 2263–2273, https://doi.org/10.1175/BAMS-D-15-00267.1, Ziliak, S. and McCloskey, D.N., 2008. The cult of statistical significance: How the standard error costs us jobs, justice, and lives. University of Michigan Press.).

We chose 1σ as this is widely used as the uncertainty of an estimated value, and use this as a way to indicate how more or less likely it is that a trend is different to zero rather than the magnitude of the trend itself. However we are actually not expecting the trends to be zero in many cases and so this has framed how we see the problem. Therefore we do not wish to add formal significance testing to this assessment.

In this study we are (a) fitting a trend to a relatively small number of points, and (b) using a linear trend to summarise changes over the 45 years of the study. We do not expect any changes to be linear, and merely use this as a way of simply quantifying changes over time, as has been done in many other studies of the past climate. Furthermore, the small number of points is the result of balancing sufficient observations per temporal bin to accurately determine the properties of a distribution with a large enough number of bins.

4, Since only data over the past 45 years were analyzed. Are the observed changes influenced by potential decadal variabilities in the climate system? Can the statistical significance test rule out the potential influences from the decadal variabilities?

As noted in the response to (3), we do not use a statistical significance test in this analysis except for the section on quantile regression. We also note that we do not expect any long-term changes to be linear, but use a linear trend to summarise these changes over time in a clear and simple way. We are in effect limited to a maximum of 46 years of data given the drop off in stations available in the HadISD prior to 1973, as discussed in Section 2. So it is quite possible that decadal variabilities could be one of the drivers for these changes in shape, and the length of data we have available may not be long enough to disentangle these effects. Our aim was to investigate what the changes in the distributions were, and point to possible causes rather than determine the most likely cause.

We have in response to comments from the reviewer 1 added discussion into the possible causes of changes in these distributions, and mention decadal variabilities there.

5, The results from this work were compared with the findings from previous studies.
When the results are not in line with each other, which results are more reliable? Why?
The authors may need to better explain why the results are different.

It is very difficult in this comparison to determine which study has the most reliable results.  The limitations with the study done here arise mostly from the amount of data available, e.g. once sub-daily records are split into hourly anomalies and then combined in 5-year periods (Section 4). However we do not have the ability to investigate the details of other studies to pull out where differences lie - which may be from the trend fitting, the data preparation, the time periods covered, the underlying stations etc.  We merely explain where our results differ from others and offer possible explanations but cannot say the exact cause or which to take.

In our re-ordering of Section 4 (from your 1st comment) we hope that we've clarified our analyses and have also added sentences outlining why we feel that we cannot say which assessment is best, especially as our opinion could come across as biased.